# Sharpin prevents skin inflammation by inhibiting TNFR1-induced keratinocyte apoptosis

Snehlata Kumari[1,2†], Younes Redouane[3†], Jaime Lopez-Mosqueda[4], Ryoko Shiraishi[3], Malgorzata Romanowska[1,2], Stefan Lutzmayer[3], Jan Kuiper[1,2], Conception Martinez[3], Ivan Dikic[4]*, Manolis Pasparakis[1,2]*, Fumiyo Ikeda[3]*

[1]Institute for Genetics, Center for Molecular Medicine, University of Cologne, Cologne, Germany; [2]Cologne Excellence Cluster on Cellular Stress Responses in Aging-Associated Diseases, University of Cologne, Cologne, Germany; [3]Institute of Molecular Biotechnology, Vienna, Austria; [4]Institute of Biochemistry II, Goethe University Medical School, Frankfurt am Main, Germany

**Abstract** Linear Ubiquitin chain Assembly Complex (LUBAC) is an E3 ligase complex that generates linear ubiquitin chains and is important for tumour necrosis factor (TNF) signaling activation. Mice lacking Sharpin, a critical subunit of LUBAC, spontaneously develop inflammatory lesions in the skin and other organs. Here we show that TNF receptor 1 (TNFR1)-associated death domain (TRADD)-dependent TNFR1 signaling in epidermal keratinocytes drives skin inflammation in Sharpin-deficient mice. Epidermis-restricted ablation of Fas-associated protein with death domain (FADD) combined with receptor-interacting protein kinase 3 (RIPK3) deficiency fully prevented skin inflammation, while single RIPK3 deficiency only delayed and partly ameliorated lesion development in Sharpin-deficient mice, showing that inflammation is primarily driven by TRADD- and FADD-dependent keratinocyte apoptosis while necroptosis plays a minor role. At the cellular level, Sharpin deficiency sensitized primary murine keratinocytes, human keratinocytes, and mouse embryonic fibroblasts to TNF-induced apoptosis. Depletion of FADD or TRADD in Sharpin-deficient HaCaT cells suppressed TNF-induced apoptosis, indicating the importance of FADD and TRADD in Sharpin-dependent anti-apoptosis signaling in keratinocytes.

*For correspondence: ivan.dikic@biochem2.de (ID); pasparakis@uni-koeln.de (MP); fumiyo.ikeda@imba.oeaw.ac.at (FI)

†These authors contributed equally to this work

## Introduction

Tumor necrosis factor receptor 1 (TNFR1)-mediated signaling is regulated by multiple ubiquitination events involving linear (Met1), Lys63 and Lys48 ubiquitination of several target proteins (*Wertz and Dixit, 2008*; *Verhelst et al., 2011*; *Schmukle and Walczak, 2012*). Upon TNF stimulation, a receptor proximal signaling complex (termed complex I) consisting of the TNFR1-associated death domain (TRADD), receptor-interacting protein kinase 1 (RIPK1), TNF receptor-associated factor 2 (TRAF2), and cellular inhibitor of apoptosis proteins 1 and 2 (cIAP1/2) is recruited to the intracellular domain of TNFR1 (*Haas et al., 2009*). Ubiquitination of proteins in complex I, including RIPK1, cIAP1/2, and TRAF2, leads to the recruitment of additional signaling components that facilitate activation of nuclear factor-κB (NF-κB) and mitogen-activated protein (MAP) kinase signaling cascades (*Wu et al., 2005*; *Varfolomeev et al., 2008*; *Haas et al., 2009*). It has been shown that the transforming growth factor beta-activated kinase 1 (TAK1)/TAK1-binding protein 2 (TAB2) complex is recruited into complex I through the interaction between the TAB2-Npl4 zinc finger (NZF) and Lys63-linked ubiquitin chains (*Kanayama et al., 2004*; *Kulathu et al., 2009*; *Sato et al., 2009*). cIAP-mediated ubiquitination of RIPK1 and cIAPs themselves was shown to result in the recruitment of the E3 ligase complex Linear

**eLife digest** In response to an injury or an infection, areas of the body can become inflamed as the immune system attempts to repair the damage and/or destroy any microbes or toxins that have entered the body. At the level of individual cells inflammation can involve cells being programmed to die in one of two ways: apoptosis and necroptosis.

Apoptosis is a highly controlled process during which the contents of the cell are safely destroyed in order to prevent damage to surrounding cells. Necroptosis, on the other hand, is not controlled: the cell bursts and releases its contents into the surroundings.

Inflammation is activated by a protein called TNFR1, which is controlled by a complex that includes a protein called Sharpin. Mice that lack the Sharpin protein develop inflammation on the skin and other organs, even in the absence of injury or infection. However, it is not clear how the Sharpin protein controls TNFR1 to prevent inflammation. Kumari et al. and, independently Rickard et al., have now studied this process in detail.

Kumari et al. have found that inflammation in mice lacking Sharpin depends on TNFR1 interacting with another protein called TRADD. The experiments also show that the inflammation is mainly driven by apoptosis, with necroptosis having only a minor role. Further experiments carried out in mammal cells showed that TRADD and another protein (called FADD) work with Sharpin to prevent apoptosis.

At the molecular level, Sharpin is known to induce a special type of protein modification (called linear ubiquitination) with two partner proteins, so the next challenge is to work out exactly how Sharpin uses this process to prevent apoptosis.

Ubiquitin chain Assembly Complex (LUBAC) (*Haas et al., 2009*) into complex I. LUBAC consists of a catalytic protein, HOIL-1L interacting protein (HOIP)/ring finger protein 31 (Rnf31), and two other critical subunits, Sharpin/Shank-interacting protein-like 1 (SIPL1) and HOIL-1L/RanBP-type and C3HC4-type zinc finger containing 1 (Rbck1) (*Gerlach et al., 2011*; *Ikeda et al., 2011*; *Tokunaga et al., 2011*). Using biochemical and cell biological approaches, LUBAC has been shown to specifically generate linear ubiquitin chains, linked via Met1, and these chain types are important for pathway activation (*Kirisako et al., 2006*). To date, LUBAC is the only E3 ligase complex identified that catalyzes linear ubiquitin chain generation. HOIP belongs to a RING-in-between-RING (RBR)-type of E3 ligase family and constitutes the catalytic center in LUBAC (*Kirisako et al., 2006*; *Stieglitz et al., 2013*). Interestingly, HOIP requires binding to either Sharpin or HOIL-1L for its catalytic action (*Kirisako et al., 2006*; *Gerlach et al., 2011*; *Ikeda et al., 2011*; *Tokunaga et al., 2011*; *Stieglitz et al., 2012*). It was shown that the LUBAC-mediated ubiquitination of NEMO in the IκB kinase (IKK) complex is critical for the NF-κB signaling pathway. Sharpin or HOIL-1L deficiency partially suppress TNFR1-induced NF-κB activation, suggesting that these components show some degree of functional redundancy in regulating NF-κB signaling (*Haas et al., 2009*; *Tokunaga et al., 2009*; *Gerlach et al., 2011*; *Ikeda et al., 2011*; *Tokunaga et al., 2011*).

Earlier studies identified Sharpin as the gene mutated in the chronic proliferative dermatitis mice (*Sharpin$^{cpdm/cpdm}$*), which spontaneously develop severe chronic inflammation primarily in the skin but also in other tissues such as the gut, lung, liver, and esophagus (*Gijbels et al., 1996*; *Seymour et al., 2007*). The pathogenesis of multi-organ chronic inflammation in *Sharpin$^{cpdm/cpdm}$* mice depends on TNF, as double *Sharpin$^{cpdm/cpdm}$*;*Tnf$^{-/-}$* mice did not develop signs of inflammatory skin and liver disease (*Gerlach et al., 2011*). These results showed that Sharpin has an essential function in preventing TNF-induced chronic inflammation. However, the molecular mechanisms that are controlled by Sharpin to prevent TNF-induced inflammatory disease remain poorly understood. Here we show that the skin inflammation in *Sharpin$^{cpdm/cpdm}$* mice is triggered by TNFR1-mediated TRADD- and FADD-dependent apoptosis of keratinocytes.

## Results

### TNFR1 deficiency in keratinocytes prevents skin inflammation in *Sharpin$^{cpdm/cpdm}$* mice

Previous studies showed that TNF is required for the development of multi-organ inflammation in *Sharpin$^{cpdm/cpdm}$* mice (*Gerlach et al., 2011*). To address whether this function of TNF is mediated by TNFR1, we crossed *Sharpin$^{cpdm/cpdm}$* mice with *Tnfrsf1a$^{-/-}$* animals. Double deficient *Sharpin$^{cpdm/cpdm}$*;*Tnfrsf1a$^{-/-}$*

mice did not develop skin inflammation, demonstrating that TNF-induced TNFR1 signaling is essential for the pathogenesis of inflammatory skin lesions in *Sharpin^cpdm/cpdm* mice (*Figure 1*). We then tried to identify the cellular target of the pathogenic TNFR1 signaling in *Sharpin^cpdm/cpdm* mice. We have recently shown that TNFR1 signaling in NF-κB-deficient epidermal keratinocytes drives psoriasis-like skin inflammation in mice (*Kumari et al., 2013*), identifying keratinocytes as an important cellular target of pathogenic TNF signaling in skin inflammation. To address whether TNFR1 signaling in epidermal keratinocytes drives the skin inflammation in *Sharpin^cpdm/cpdm* mice, we crossed *Sharpin^cpdm/cpdm* mice with *K14Cre-Tnfrsf1a^fl/fl* (TNFR1^E-KO) mice that lack TNFR1 specifically in keratinocytes (*Figure 1A*). These *Sharpin^cpdm/cpdm*;TNFR1^E-KO mice did not develop any macroscopic signs of skin inflammation (*Figure 1B*). In addition, histological analysis of *Sharpin^cpdm/cpdm*;TNFR1^E-KO mice skin revealed a normal epidermis without keratinocyte death (cleaved caspase-3 staining in *Figure 1C*), skin inflammation (F4/80 staining in *Figure 1C*), or epidermal hyperplasia (H&E, Keratin 6, Keratin 10, and Loricrin staining in *Figure 1C* and quantification in *Figure 1D*), similar to *Sharpin^cpdm/cpdm*;*Tnfrsf1a* ^−/− animals (*Figure 1C,D*). These results demonstrate that TNFR1 signaling in epidermal keratinocytes is essential for the pathogenesis of skin inflammation in *Sharpin^cpdm/cpdm* mice.

## Epidermal FADD-dependent TNFR1-induced death of keratinocytes induces skin inflammation in *Sharpin^cpdm/cpdm* mice

Having established keratinocyte-intrinsic TNFR1 signaling as a key spatial event triggering skin inflammation in *Sharpin^cpdm/cpdm* mice, we sought to investigate the cell death mechanisms by which epithelial TNFR1 induces the inflammatory response. We and others have shown that increased numbers of keratinocytes in the epidermis of *Sharpin^cpdm/cpdm* mice undergo apoptosis, as indicated by the presence of cleaved caspase-3 (*Ikeda et al., 2011*; *Liang and Sundberg, 2011*) (also see *Figure 1C*). In addition, it was suggested that Sharpin deficiency sensitizes primary keratinocytes to both TNF-induced caspase-dependent apoptosis and RIP-kinase-dependent necroptosis (*Gerlach et al., 2011*). We therefore used genetic mouse models to address the role of FADD/caspase-8-dependent apoptosis and RIPK3-dependent necroptosis in *Sharpin^cpdm/cpdm* mice. To address the role of RIPK3-dependent necroptosis, we generated mice lacking both Sharpin and RIPK3 by crossing *Sharpin^cpdm/cpdm* with *Ripk3^−/−* animals (*Figure 2*). Double deficient *Sharpin^cpdm/cpdm*;*Ripk3^−/−* mice developed skin lesions similar to those of *Sharpin^cpdm/cpdm* mice, demonstrating that RIPK3 deficiency did not prevent the development of skin inflammation (*Figure 2A,B*). However, the initiation of the skin phenotype was delayed in *Sharpin^cpdm/cpdm*;*Ripk3^−/−* animals, which started to show lesions after the age of 10 weeks but showed a large variability in onset and severity with some mice showing only mild lesions even at the age of 19 weeks (*Figure 2—figure supplement 1*). *Sharpin^cpdm/cpdm* mice also showed variability with lesion onset between 8 and 11 weeks, but all mice showed severe lesions by the age of 12–14 weeks. Quantification of epidermal thickness revealed that RIPK3 deficiency mildly ameliorated the severity of skin lesions (*Figure 2C*). These results showed that, although RIPK3-dependent necroptosis contributes to accelerating the onset and exacerbating the severity of the phenotype, it is not essential for the pathogenesis of the inflammatory skin lesions in *Sharpin^cpdm/cpdm* mice.

Next, we sought to address whether FADD/caspase-8-dependent apoptosis of Sharpin-deficient keratinocytes induces TNF-dependent skin inflammation in *Sharpin^cpdm/cpdm* mice. Since deficiency of caspase-8 or FADD alone in epidermal keratinocytes triggers a RIPK3-dependent skin inflammation (*Bonnet et al., 2011*; *Weinlich et al., 2013*), we could not directly investigate the role of FADD or caspase-8 in the TNF-induced death of Sharpin-deficient keratinocytes in vivo. However, taking advantage of the fact that concomitant deletion of RIPK3 fully prevents skin lesion formation in mice with keratinocyte-restricted FADD knockout (FADD^E-KO) (*Bonnet et al., 2011*), we generated *Sharpin^cpdm/cpdm*;*Ripk3^−/−* mice that also lacked FADD specifically in keratinocytes (*Figure 2*). These *Sharpin^cpdm/cpdm*;FADD^E-KO;*Ripk3^−/−* mice did not develop any macroscopic (*Figure 2A*) or histological (*Figure 2B*) skin lesions up to the age of 4 months, as they showed normal keratinocyte proliferation and differentiation without any signs of inflammation or epidermal hyperplasia (*Figure 2B,C*). Moreover, apoptosis of keratinocytes observed in *Sharpin^cpdm/cpdm* mice was completely prevented in *Sharpin^cpdm/cpdm*;FADD^E-KO;*Ripk3^−/−* mice, as shown by the absence of cleaved caspase-3 positive cells (compare *Figure 2B* with *Figure 1C*). Taken together, these results showed that combined inhibition of FADD/caspase-8-dependent apoptosis and RIPK3-dependent necroptosis prevented keratinocyte death and the development of skin lesions in *Sharpin^cpdm/cpdm* mice, providing in vivo genetic evidence that skin inflammation is triggered by TNFR1-induced death of Sharpin-deficient keratinocytes.

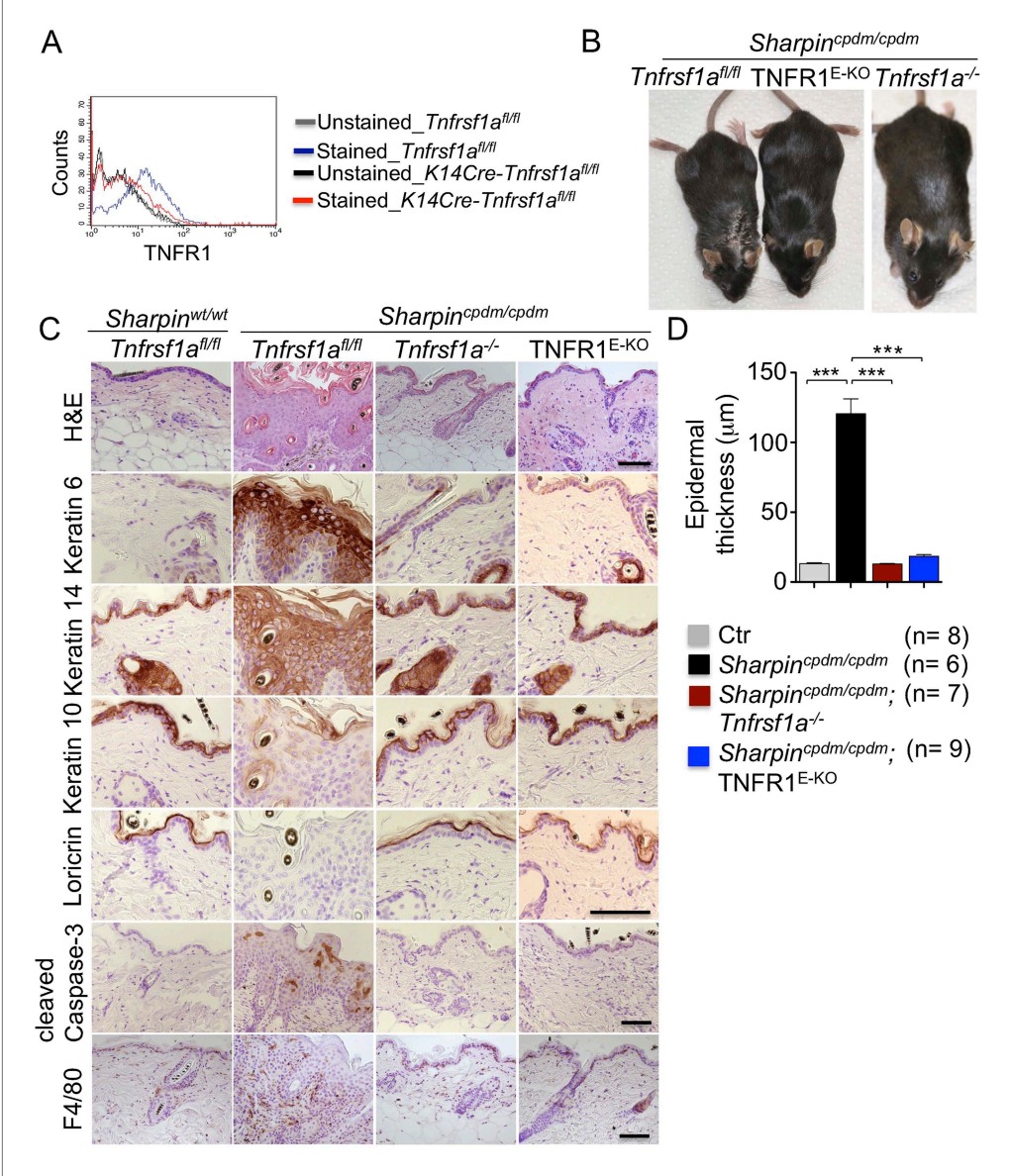

**Figure 1**. Tumor necrosis factor receptor 1 (TNFR1) signaling in keratinocytes triggers chronic proliferative dermatitis phenotype in *Sharpin^cpdm/cpdm* mice. (**A**) Flow cytometric analysis of TNFR1 expression on the isolated keratinocytes from mice with the indicated genotypes. (**B** and **C**) Macroscopic pictures, Hematoxylin and Eosin staining (H&E), Keratin 6, 14, 10 and Loricrin as well as cleaved caspase-3 and F4/80 staining of the skin sections from 14-week-old littermate mice of the indicated genotypes. The scale bars are 100 μm. (**D**) Microscopic quantification of the epidermal thickness from 12–18-week-old mice of the indicated genotypes and their littermate controls (Ctr), which consisted of the following genotypes: *Sharpin^cpdm/wt*;*Tnfrsf1a^–/–*, *Sharpin^wt/wt*;*Tnfrsf1a^–/–*, *Sharpin^cpdm/wt*;*Tnfrsf1a^fl/fl*, *Sharpin^cpdm/wt*;TNFR1^E-KO, and *Sharpin^wt/wt*;TNFR1^E-KO. The *Sharpin^cpdm/cpdm* group consisted of *Sharpin^cpdm/cpdm*;*Tnfrsf1a^fl/fl* and *Sharpin^cpdm/cpdm*;*Tnfrsf1a^fl/wt* mice that were littermates of the *Sharpin^cpdm/cpdm*;TNFR1^E-KO mice. The *Sharpin^cpdm/cpdm*;*Tnfrsf1a^–/–* mice were derived from a different line and shown here is the picture and the staining from the age-matched mice. Bars represent mean values ± SEM. Statistical significance was determined using the Student's *t* test (***p ≤ 0.001).

## TRADD-dependent apoptosis of Sharpin-deficient keratinocytes triggers skin inflammation in *Sharpin^cpdm/cpdm* mice

Our findings strongly suggest that FADD-dependent apoptosis of Sharpin-deficient keratinocytes triggers skin inflammation. However, since the role of FADD can only be addressed in a RIPK3-deficient

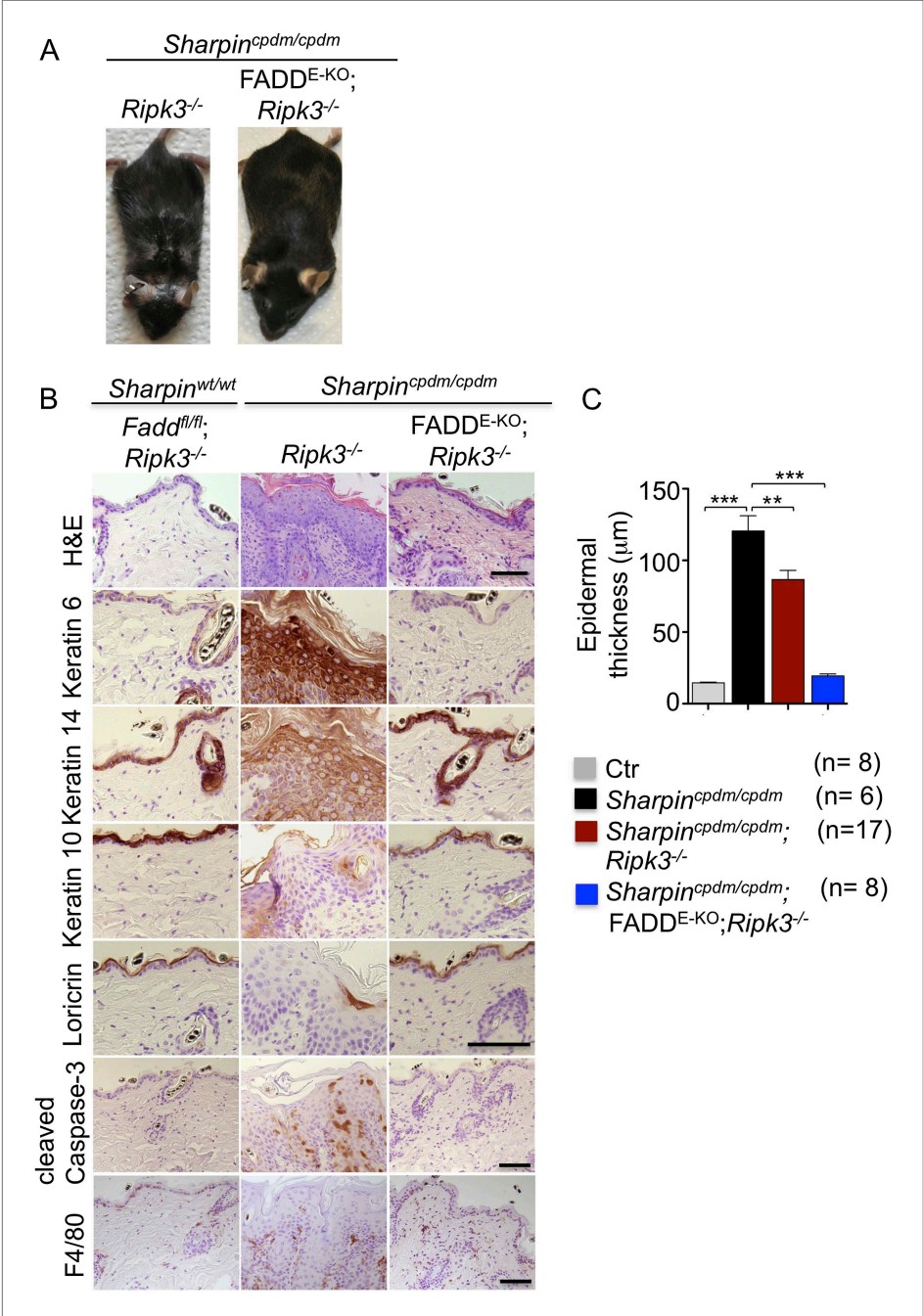

**Figure 2**. Fas-associated protein with death domain (FADD) deficiency in keratinocytes prevents skin inflammation in *Sharpin^cpdm/cpdm* mice. (**A** and **B**) Macroscopic pictures, Hematoxylin and Eosin staining (H&E), Keratin 6, 14, 10 and Loricrin as well as cleaved caspase-3 and F4/80 staining of skin sections from 14-week-old mice of the indicated genotypes. The scale bars are 100 μm. (**C**) Microscopic quantification of the epidermal thickness from 12–18-week-old mice of the indicated genotypes and their littermate controls (Ctr), which consisted of the following genotypes: *Sharpin^wt/wt*;*Tnfrsf1a^fl/fl*, *Sharpin^cpdm/wt*;*Fadd^fl/fl*;*Ripk3^−/−*, *Sharpin^wt/wt*;*Fadd^fl/fl*;*Ripk3^−/−*, *Sharpin^cpdm/wt*;FADD^E-KO^;*Ripk3^−/−* and *Sharpin^wt/wt*;FADD^E-KO^;*Ripk3^−/−*. The *Sharpin^cpdm/cpdm* group consisted of *Sharpin^cpdm/cpdm*;*Tnfrsf1a^fl/fl* and *Sharpin^cpdm/cpdm*;*Tnfrsf1a^fl/wt* mice that were littermates of the *Sharpin^cpdm/cpdm*;TNFR1^E-KO^ mice. *Sharpin^cpdm/cpdm*; *Ripk3^−/−* mice were derived from the same breeding line as *Sharpin^cpdmcpdm*;FADD^E-KO^;*Ripk3^−/−* mice and consisted of the genotype *Sharpin^cpdm/cpdm*;*Fadd^fl/fl*;*Ripk3^−/−*. Bars represent mean values ± SEM. Statistical significance was determined using the Student's *t* test (***p ≤ 0.001, **p ≤ 0.01).

*Figure 2. Continued on next page*

*Figure 2. Continued*

The following figure supplement is available for figure 2:

**Figure supplement 1**. Variability among *Sharpin^cpdm/cpdm*;*Ripk3^−/−* mice at a similar age.

background, it remains possible that FADD-dependent apoptosis and RIPK3-dependent necroptosis might share a redundant function in inducing the cell death of Sharpin-deficient keratinocytes and triggering skin inflammation. To directly address the role of TNFR1-induced apoptosis in *Sharpin^cpdm/cpdm* mice, we employed mice carrying conditional alleles for TRADD, an adapter molecule that is important for the induction of inflammatory and apoptotic signaling downstream of TNFR1 (*Chen et al., 2008*; *Michallet et al., 2008*). It has been shown that TRADD deficiency partially inhibits TNFR1-induced activation of NF-κB and MAP kinase pathways and fully prevents TNFR1-induced apoptosis in mouse embryonic fibroblasts (MEFs) in vitro and in hepatocytes in vivo (*Ermolaeva et al., 2008*). To examine the role of TRADD in TNFR1-induced apoptosis and necroptosis, we analyzed the response of wild type and TRADD-deficient primary MEFs to TNF stimulation in the presence of cycloheximide (CHX), caspase inhibitor (Z-VAD-FMK), and RIPK1 inhibitor (Necrostatin-1) (*Figure 3A*). As expected, TRADD-deficient MEFs were resistant to apoptosis induced by TNF and CHX. However, in contrast to earlier studies (*Pobezinskaya et al., 2008*), we found that TRADD-deficient MEFs were sensitive to necroptosis induced by TNF, CHX, and Z-VAD-FMK. These results demonstrated that TRADD deficiency specifically blocks TNFR1-induced apoptosis (*Figure 3A*). We therefore generated *Sharpin^cpdm/cpdm* mice lacking TRADD specifically in keratinocytes by crossing *Sharpin^cpdm/cpdm* with *K14Cre-Tradd^fl/fl* mice (*Sharpin^cpdm/cpdm*; TRADD^E-KO) (*Figure 3B–D*). Indeed, keratinocyte-restricted TRADD deficiency prevented skin lesion development in *Sharpin^cpdm/cpdm* mice, as shown by macroscopic and histological analysis (*Figure 3B–D*). Collectively, our results show that TNFR1-induced TRADD- and FADD-dependent apoptosis of Sharpin-deficient keratinocytes triggers the chronic proliferative dermatitis phenotype in *Sharpin^cpdm/cpdm* mice.

## TNFR1-induced signaling in non-epidermal cells induces extracutaneous organ inflammation in *Sharpin^cpdm/cpdm* mice

In addition to dermatitis, *Sharpin^cpdm/cpdm* mice develop splenomegaly and inflammation in other organs such as liver and lung (*Figure 4A,B*). We found that systemic deficiency of TNFR1 prevented the development of liver inflammation in *Sharpin^cpdm/cpdm* mice (*Figure 4A*), consistent with earlier results showing that TNF deficiency also inhibited liver inflammation in these animals (*Gerlach et al., 2011*). We also found that TNFR1 deficiency prevented lung inflammation (*Figure 4A*) and the development of splenomegaly (*Figure 4B*) in *Sharpin^cpdm/cpdm* mice. TNFR1 deficiency also corrected the splenic structure defects in *Sharpin^cpdm/cpdm* mice (*Figure 4A*), in contrast to the report by *Gerlach et al. (2011)* that TNF deficiency could not rescue the splenic structural abnormalities of *Sharpin^cpdm/cpdm* mice. This contradiction likely stems from the fact that Gerlach et al. compared the spleens of *Sharpin^cpdm/cpdm* mice with wild type mice. However, considering that TNFR1- and TNF-deficient animals have altered splenic structures characterized by lack of B cell lymphoid follicles and marginal zone abnormalities (*Pasparakis et al., 1996a*, *1996b*, *2000*), TNFR1 or TNF deficiency cannot restore the spleen structure of *Sharpin^cpdm/cpdm* mice to that of a wild type mouse. Keratinocyte-restricted TNFR1 deficiency could not rescue the extracutaneous pathologies in *Sharpin^cpdm/cpdm* mice, suggesting that these develop independently from the skin lesions (*Figure 4A,B*). These results demonstrate that TNFR1 signaling in non-epidermal cells triggers splenomegaly and extracutaneous inflammation in *Sharpin^cpdm/cpdm* mice. We also analyzed the liver, lung, and spleen of *Sharpin^cpdm/cpdm*;*Ripk3^−/−* and *Sharpin^cpdm/cpdm*;FADD^E-KO; *Ripk3^−/−* mice. Interestingly, we observed partial normalization of liver and lung inflammation (*Figure 5A*), splenomegaly and the splenic structure in *Sharpin^cpdm/cpdm*;*Ripk3^−/−* mice (*Figure 5A,B*), suggesting that RIPK3-mediated necroptosis contributes to the extracutaneous inflammatory pathologies. *Sharpin^cpdm/cpdm*; FADD^E-KO;*Ripk3^−/−* mice showed very similar histology of the liver, lung, and spleen to *Sharpin^cpdm/cpdm*; *Ripk3^−/−* mice, indicating that the extracutaneous phenotype in *Sharpin^cpdm/cpdm*;FADD^E-KO;*Ripk3^−/−* mice is mainly attributed to RIPK3 signaling and not to epidermal FADD signaling (*Figure 5A*). In addition, we performed cleaved caspase-3 staining and TUNEL staining on liver, lung, and spleen tissue sections obtained from *Sharpin^cpdm/cpdm*, *Sharpin^cpdm/cpdm*;*Ripk3^−/−*, *Sharpin^cpdm/cpdm*;*Tnfrsf1a^−/−*, *Sharpin^cpdm/cpdm*;TNFR1^E-KO, and *Sharpin^cpdm/cpdm*;FADD^E-KO;*Ripk3^−/−* mice and from their respective controls *Sharpin^cpdm/wt*, *Sharpin^cpdm/wt*;*Ripk3^−/−*, *Sharpin^cpdm/wt*;*Tnfrsf1a^−/−*, *Sharpin^cpdm/cpdm*;TNFR1^E-KO, and *Sharpin^cpdm/wt*;

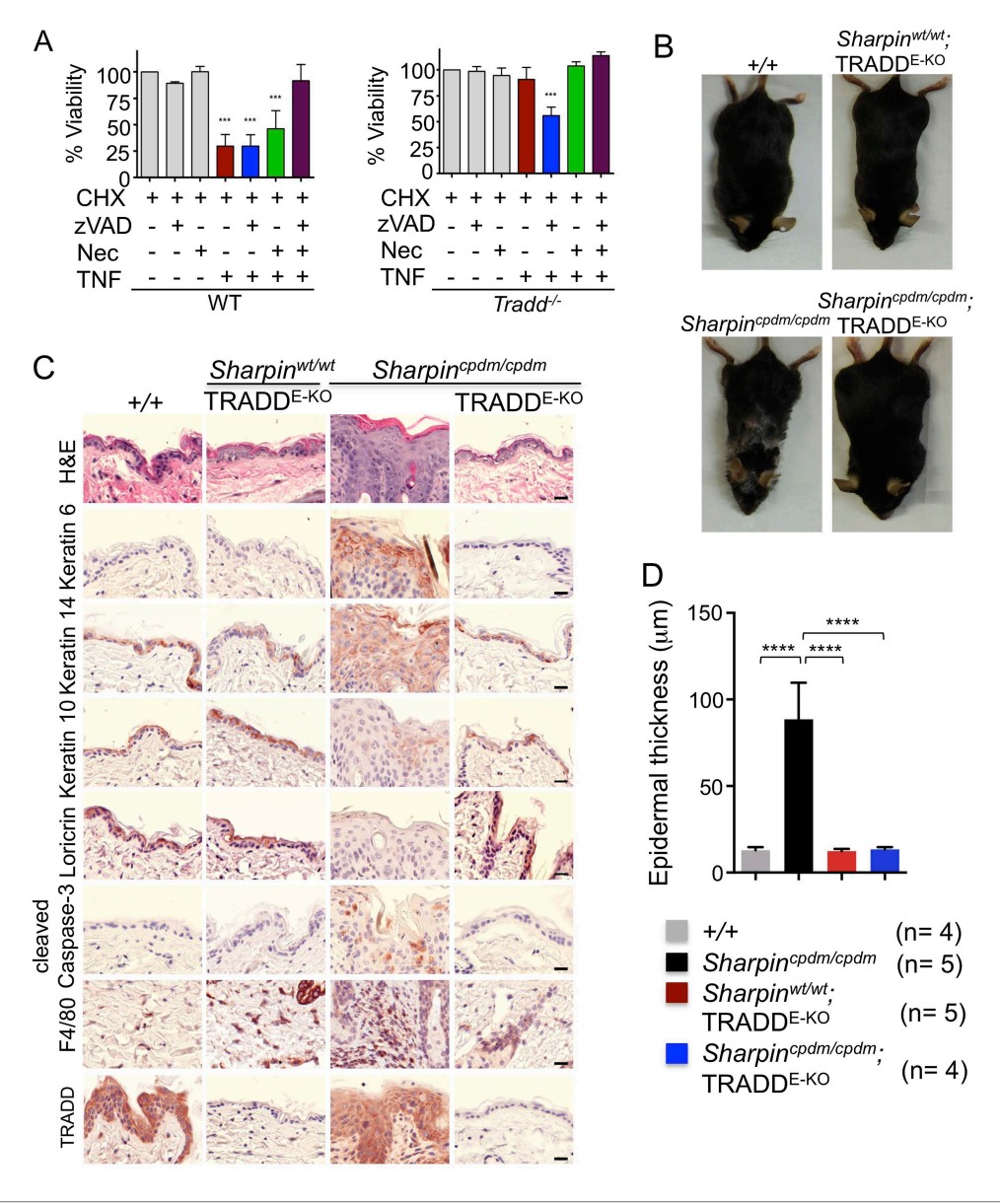

**Figure 3**. Tumor necrosis factor receptor 1-associated death domain (TRADD) deficiency in keratinocytes prevents skin inflammation in *Sharpin^{cpdm/cpdm}* mice. (**A**) The percentage viability of wild type (WT) mouse embryonic fibroblasts (MEFs) (n = 3) and TRADD-deficient MEFs (n = 3) upon tumor necrosis factor (TNF), cycloheximide (CHX), caspase inhibitor (zVAD), and Necrostatin-1 (Nec) treatment alone or in combination for 20 hr and measurement by WST-1 assay. Bars represent average cell viability (± SD) of three independent experiments. (**B** and **C**) Macroscopic gross appearance of the WT (+/+) and littermate mice of the indicated genotype at the age of 12 weeks (**B**) and (H&E), Keratin 6, 14, 10 and Loricrin as well as cleaved caspase-3 and F4/80 staining of the skin sections from 12-week-old mice of the indicated genotypes (**C**). Scale bars in (**C**) are 100 μm. (**D**) Microscopic quantification of the epidermal thickness from 12-week-old littermate mice of *Sharpin^{cpdm/cpdm}*, *Sharpin^{wt/wt}*;TRADD^{E-KO}, *Sharpin^{cpdm/cpdm}*;TRADD^{E-KO} and age-matched WT (+/+) is shown. Bars represent mean values ± SD. Statistical significance was determined using the one-way ANOVA test (****p ≤ 0.0001).

FADD^{E-KO};*Ripk3^{−/−}* mice (n = 2–4 mice from each genotype) (*Figure 5—figure supplements 1–3*). We observed increased active caspase-3 and TUNEL positive cells in the lung and liver of *Sharpin^{cpdm/cpdm}* mice compared with *Sharpin^{cpdm/wt}* mice, whereas the spleen did not seem to have an increased number of cleaved caspase-3 positive cells compared with the control mice. The numbers of cleaved caspase-3

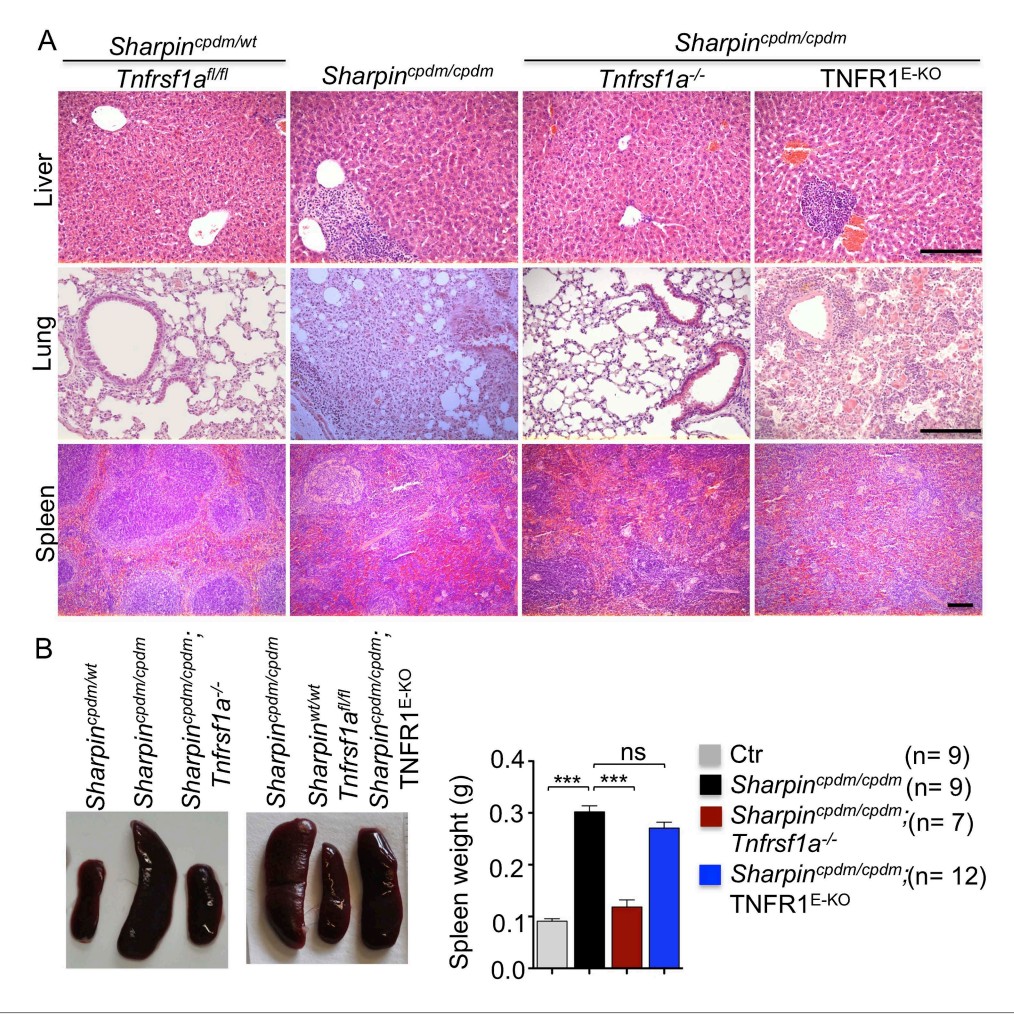

**Figure 4**. Tumor necrosis factor receptor 1 (TNFR1) deficiency in *Sharpin^cpdm/cpdm* mice rescues the inflammation of lung, liver and splenomegaly but not epidermal keratinocyte-restricted knockout of TNFR1. (**A** and **B**) H&E staining of liver, lung and spleen, and macroscopic pictures of spleen from mice with the indicated genotypes as well as measurement of spleen weight from 12–18-week-old *Sharpin^cpdm/cpdm*, *Sharpin^cpdm/cpdm*;*Tnfrsf1a^−/−*, and *Sharpin^cpdm/cpdm*; TNFR1^E-KO mice and their littermate controls (Ctr), which consisted of the following genotypes: *Sharpin^cpdm/wt*; *Tnfrsf1a^−/−*, *Sharpin^wt/wt*;*Tnfrsf1a^−/−*, *Sharpin^cpdm/wt*;*Tnfrsf1a^fl/fl*, *Sharpin^cpdm/wt*;TNFR1^E-KO, and *Sharpin^wt/wt*;TNFR1^E-KO. The *Sharpin^cpdm/cpdm* group consisted of *Sharpin^cpdm/cpdm*;*Tnfrsf1a^fl/fl* and *Sharpin^cpdm/cpdm*;*Tnfrsf1a^fl/wt* mice that were littermates of the *Sharpin^cpdm/cpdm*;TNFR1^E-KO mice. Scale bars in (**A**) are 100 μm. Results are expressed as mean values ± SEM. Statistical significance was determined using unpaired Student's *t* test (two-tailed) (***p ≤ 0.001).

positive and TUNEL positive cells were reduced in *Sharpin^cpdm/cpdm*;*Ripk3^−/−*, *Sharpin^cpdm/cpdm*;FADD^E-KO; *Ripk3^−/−* and *Sharpin^cpdm/cpdm*;*Tnfrsf1a^−/−* mice, whereas *Sharpin^cpdm/cpdm*;TNFR1^E-KO mice showed similar cleaved caspase-3 and TUNEL positive cells to *Sharpin^cpdm/cpdm* mice. These findings essentially showed that, in the mice showing no or less inflammation, the numbers of both cleaved caspase-3 and TUNEL positive cells were reduced. Furthermore, we observed that liver and lung inflammation as well as splenomegaly and the defect of splenic structure were not altered by keratinocyte-specific depletion of TRADD in *Sharpin^cpdm/cpdm* mice (*Figure 6A,B*), providing further support for the notion that the development of extracutaneous organ inflammation in *Sharpin^cpdm/cpdm* mice occurs independently from the skin lesions.

## Sharpin regulates TNF-induced apoptosis

Our in vivo genetic experiments showed that TNFR1-induced apoptosis of *Sharpin^cpdm/cpdm* keratinocytes causes skin inflammation in *Sharpin^cpdm/cpdm* mice. We therefore investigated the molecular

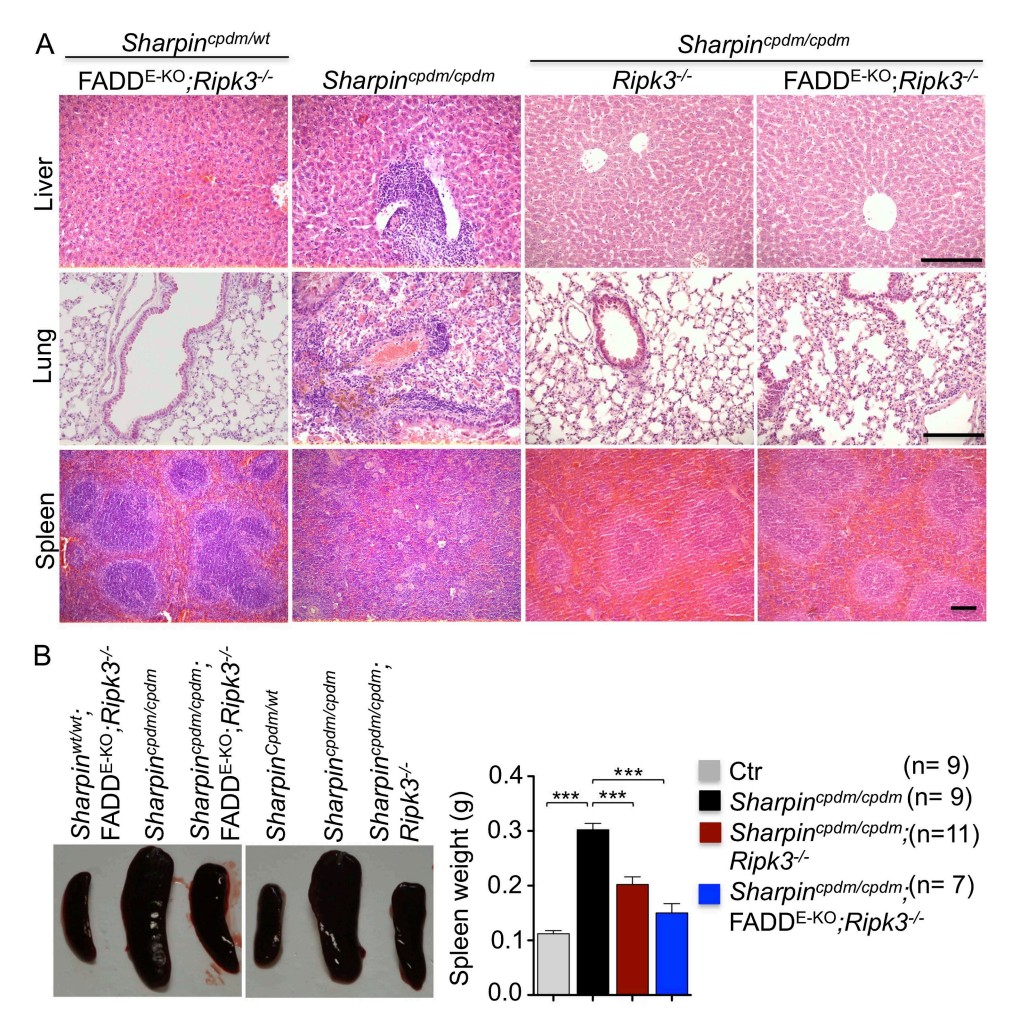

**Figure 5**. Receptor-interacting protein kinase 3 (*Ripk3*[−/−]) and epidermis-specific Fas-associated protein with death domain (FADD) together with *Ripk3*[−/−] (FADD[E-KO];*Ripk3*[−/−]) in *Sharpin*[cpdm/cpdm] mice partially rescues the inflammation of the lung, liver, and splenomegaly. (**A** and **B**) H&E staining of liver, lung and spleen, and macroscopic pictures of spleen from mice with the indicated genotypes as well as measurement of spleen weight from 12–18-week-old *Sharpin*[cpdm/cpdm], *Sharpin*[cpdm/cpdm];*Ripk3*[−/−] and *Sharpin*[cpdm/cpdm];FADD[E-KO];*Ripk3*[−/−] mice and their littermate controls (Ctr), which consisted of the following genotypes: *Sharpin*[wt/wt];*Tnfrsf1a*[fl/fl], *Sharpin*[cpdm/wt];*Fadd*[fl/fl];*Ripk3*[−/−], *Sharpin*[wt/wt]; *Fadd*[fl/fl];*Ripk3*[−/−], *Sharpin*[cpdm/wt];FADD[E-KO];*Ripk3*[−/−], and *Sharpin*[wt/wt];FADD[E-KO];*Ripk3*[−/−]. The *Sharpin*[cpdm/cpdm] group consisted of *Sharpin*[cpdm/cpdm];*Tnfrsf1a*[fl/fl] and *Sharpin*[cpdm/cpdm];*Tnfrsf1a*[fl/wt] mice that were littermates of the *Sharpin*[cpdm/cpdm];TNFR1[E-KO] mice. The *Sharpin*[cpdm/cpdm];*Ripk3*[−/−] mice were derived from the same breeding line as *Sharpin*[cpdmcpdm];FADD[E-KO];*Ripk3*[−/−] mice and consisted of the genotype *Sharpin*[cpdm/cpdm];*Fadd*[fl/fl];*Ripk3*[−/−]. Scale bars in (**A**) are 100 μm. Results are expressed as mean values ± SEM. Statistical significance was determined using unpaired Student's *t* test (two-tailed) (***p ≤ 0.001).

The following figure supplements are available for figure 5:

**Figure supplement 1**. Cell death in the spleen of *Sharpin*[cpdm/cpdm], *Sharpin*[cpdm/cpdm]; *Tnfrsf1a*[−/−], *Sharpin*[cpdm/cpdm]; TNFR1[E-KO], *Sharpin*[cpdm/cpdm];*Ripk3*[−/−] and *Sharpin*[cpdm/cpdm];FADD[E-KO];*Ripk3*[−/−] mice.

**Figure supplement 2**. Cell death in the liver of *Sharpin*[cpdm/cpdm], *Sharpin*[cpdm/cpdm]; *Tnfrsf1a*[−/−], *Sharpin*[cpdm/cpdm]; TNFR1[E-KO], *Sharpin*[cpdm/cpdm];*Ripk3*[−/−] and *Sharpin*[cpdm/cpdm];FADD[E-KO];*Ripk3*[−/−] mice.

**Figure supplement 3**. Cell death in the lung of *Sharpin*[cpdm/cpdm], *Sharpin*[cpdm/cpdm]; *Tnfrsf1a*[−/−], *Sharpin*[cpdm/cpdm]; TNFR1[E-KO], *Sharpin*[cpdm/cpdm];*Ripk3*[−/−] and *Sharpin*[cpdm/cpdm];FADD[E-KO];*Ripk3*[−/−] mice.

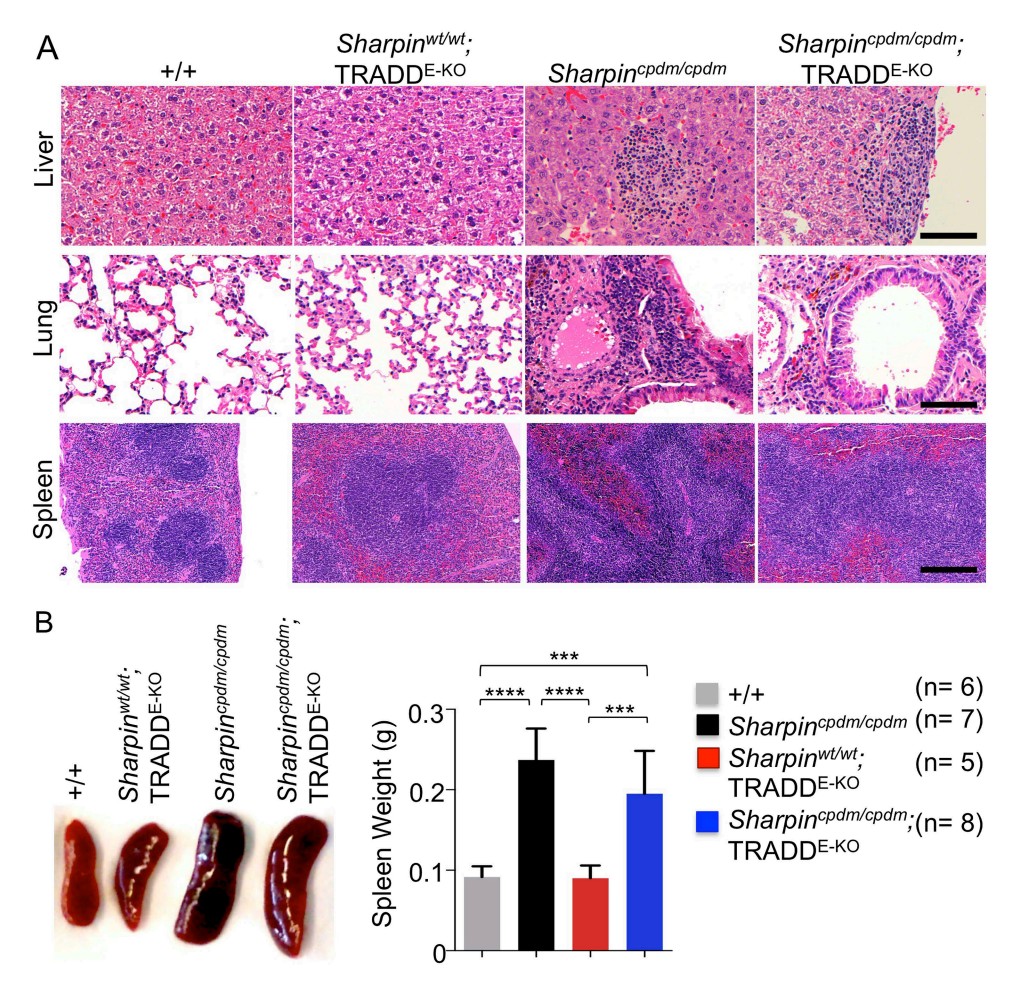

**Figure 6**. Epidermal keratinocyte-restricted knockout of tumor necrosis factor receptor 1-associated death domain (TRADD) (TRADD[E-KO]) in *Sharpin*[cpdm/cpdm] mice has a minor effect on the inflammation of the lung, liver, and spleno-megaly. (**A** and **B**) H&E staining of liver, lung, and spleen and macroscopic pictures of spleen as well as measurement of spleen weight from mice with the indicated genotypes derived from wild type (+/+), *Sharpin*[cpdm/cpdm], TRADD[E-KO] or *Sharpin*[cpdm/cpdm];TRADD[E-KO] mice at the age of 12 weeks. The mice used here are the littermates. Scale bars in (**A**) are 100 μm. Results are expressed as mean values ± SD. Statistical significance was determined using ANOVA test (***p ≤ 0.001 and ****p ≤ 0.0001).

mechanisms by which Sharpin prevents TNFR1-induced apoptosis. We first examined the induction of apoptosis in MEFs derived from *Sharpin*[cpdm/cpdm] mice stimulated with TNF alone or TNF + CHX using various assays. We observed increased sensitivity to TNF + CHX-induced apoptosis in *Sharpin*[cpdm/cpdm] MEFs compared with wild type MEFs, as determined by immunocytochemical detection of cleaved caspase-3 (*Figure 7A*), detection of annexin V positive cells by fluorescence-activated cell sorting (FACS) analysis (*Figure 7B*), analysis of the cleavage of caspase-3 and Poly (ADP-ribose) polymerase (PARP) by immunoblotting (*Figure 7C*) (*Ikeda et al., 2011*), and luminescent-based caspase-8 activity assay (*Figure 7D*). *Sharpin*[cpdm/cpdm] MEFs also showed increased cleavage of caspase-3 and PARP (*Figure 7E*) and caspase-8 activity (*Figure 7F*) in response to stimulation with TNF alone although, in the absence of CHX, the response was considerably weaker. To further analyze the cellular functions of Sharpin in anti-apoptotic signaling, we used a human keratinocyte cell line, HaCaT cells (*Boukamp et al., 1988*). By using lentiviral-based shRNA knockdown, Sharpin was stably depleted in HaCaT cells (*Figure 8A*). Sharpin-deficient HaCaT and control shRNA-introduced HaCaT cells were treated with TNF alone or TNF + CHX, and the induction of apoptosis was assessed by FACS analysis of annexin V positive cells (*Figure 8B*) and measurement of caspase-8 activity (*Figure 8C*). Similar to *Sharpin*[cpdm/cpdm]

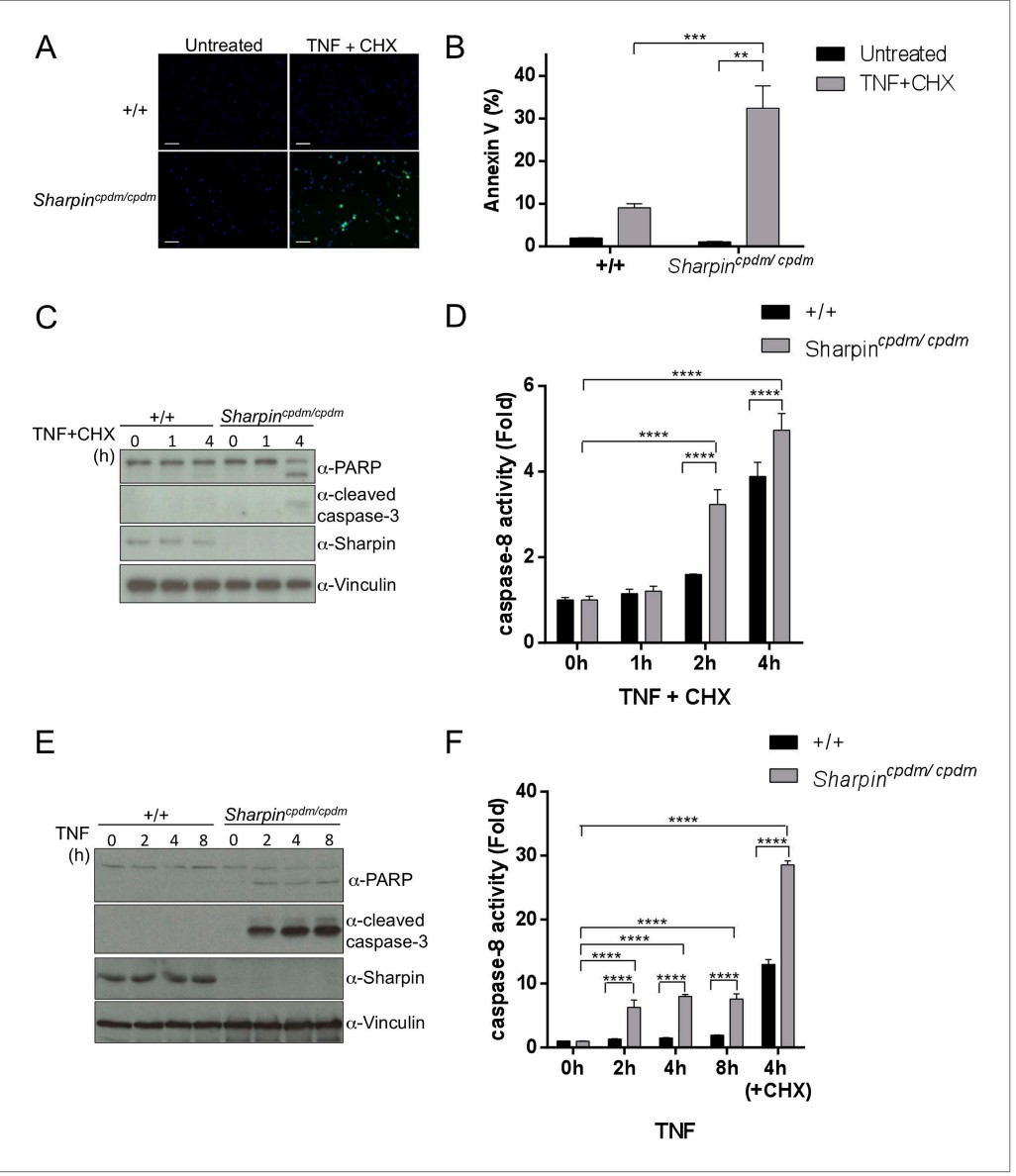

**Figure 7**. Sharpin regulates tumor necrosis factor (TNF)-induced apoptosis signaling cascade in mouse embryonic fibroblasts (MEFs). (**A**–**D**) TNF- and cycloheximide (CHX)-induced apoptosis in wild type (+/+) or *Sharpin^cpdm/cpdm* MEFs. Apoptosis in MEFs stimulated with TNF (10 ng/ml) and CHX (1 µg/ml) for 4 hr was examined by immunofluorescent staining using α-cleaved caspase-3 antibody with Alexa488 conjugated secondary antibody (**A**), by fluorescence-activated cell sorting (FACS) analysis using annexin V staining (**B**), by immunoblotting using α-cleaved caspase-3 and α-Poly (ADP-ribose) polymerase (PARP) antibodies (**C**), or by caspase-8 activity assay measured using a luminol-based assay. Scale bars in (**A**) are 100 µm. (**E** and **F**) TNF-induced apoptosis in MEFs analyzed by immunoblotting (**E**) or by caspase-8 activity assay (**F**) as in (**C** and **D**). Results are expressed as mean values ± SD. Statistical significance was determined using ANOVA test (*p ≤ 0.05, **p ≤ 0.01, ***p ≤ 0.001, ****p ≤ 0.0001).

MEFs, Sharpin knockdown sensitized HaCaT cells to apoptosis induced by TNF or TNF + CHX. Interestingly, treatment with Necrostatin-1, a RIPK1 inhibitor, suppressed caspase-8 activation in Sharpin-deficient HaCaT cells (*Figure 8D*), suggesting an involvement of RIPK1 in the induction of TNF-induced death of Sharpin-deficient keratinocytes as suggested recently by Berger et al. (*Berger et al., 2014*). Our results collectively suggest that Sharpin plays a critical role in protecting against TNF-induced apoptosis. To distinguish between the possibilities that the anti-apoptotic protection afforded by Sharpin is a LUBAC-independent function of Sharpin, we assessed the involvement of the catalytic LUBAC

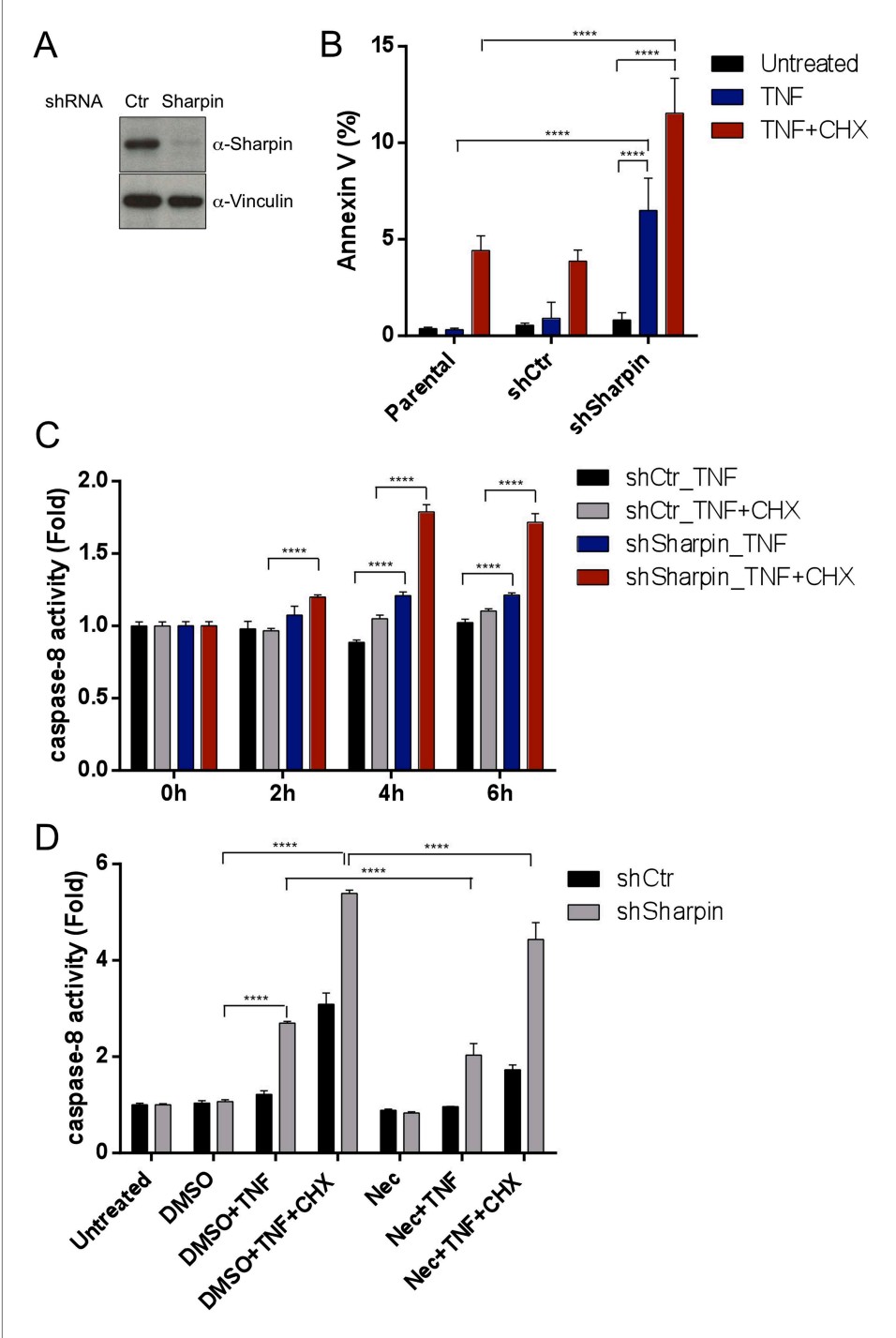

**Figure 8**. Sharpin regulates tumor necrosis factor (TNF)-induced apoptosis signaling cascade in HaCaT cells.
(**A**) Immunoblotting of stable knockdown Sharpin in HaCaT cells using α-Sharpin antibody. Control shRNA (Ctr) was used for the control knockdown. α-Vinculin antibody was used for the loading control. (**B**) Fluorescence-activated cell sorting (FACS) analysis of annexin V staining in parental, Ctr, and Sharpin knockdown HaCaT cells, stimulated with TNF (100 ng/ml) for 16 hr or TNF with cycloheximide (CHX) (1 µg/ml) for 6 hr. (**C**) Caspase-8 activity measurement using a luminol-based assay upon stimulation with TNF alone, or TNF and CHX for the indicated time in Ctr and Sharpin knockdown HaCaT cells. (**D**) Caspase-8 activity measurement upon stimulation with TNF, TNF + CHX with or without Necrostatin-1 (Nec) (30 µmol) for 6 hr in Ctr and Sharpin knockdown HaCaT cells. Results are expressed as mean values ± SD. Statistical significance was determined using ANOVA test (****p ≤ 0.0001).

*Figure 8. Continued on next page*

*Figure 8. Continued*

The following figure supplement is available for figure 8:

**Figure supplement 1**. A LUBAC component, HOIL-1L interacting protein (HOIP), plays a role in tumor necrosis factor (TNF)-induced apoptosis in HaCaT cells.

component HOIP in the regulation of apoptosis. HOIP expression was stably knocked down in HaCaT cells using shRNA (*Figure 8—figure supplement 1A*), and the cells were examined for apoptosis induced by TNF alone or TNF + CHX using FACS analysis (*Figure 8—figure supplement 1B*) and a caspase-8 activity assay (*Figure 8—figure supplement 1C and D*). As expected and consistent with a LUBAC-dependent Sharpin function, HOIP deficiency sensitized HaCaT cells to apoptosis induced by TNF alone or by TNF + CHX, akin to that observed in *Sharpin$^{cpdm/cpdm}$* MEFs or Sharpin-deficient HaCaT cells. There results provide strong support that a LUBAC-dependent Sharpin function plays a role in the regulation of apoptosis in keratinocytes.

## FADD and TRADD have a major role in Sharpin-regulated TNF-induced apoptosis

As keratinocyte apoptosis and skin inflammation in *Sharpin$^{cpdm/cpdm}$* mice were suppressed by epidermal-specific deletion of FADD or TRADD, we sought to investigate how the lack of FADD, TRADD, and RIPK3 proteins impact on TNF-induced cell viability in Sharpin-deficient cells. To address this, primary keratinocytes were isolated from newborn pups, *Sharpin$^{cpdm/cpdm}$*, *Sharpin$^{cpdm/cpdm}$*; *Ripk3$^{-/-}$*, *Sharpin$^{cpdm/cpdm}$*;FADD$^{E-KO}$;*Ripk3$^{-/-}$*, and their control littermates, *Sharpin$^{cpdm/wt}$* or *Sharpin$^{wt/wt}$* *Sharpin$^{cpdm/wt}$*;*Ripk3$^{-/-}$* and *Sharpin$^{cpdm/wt}$*;FADD$^{E-KO}$;*Ripk3$^{-/-}$*, respectively. Cells were treated with increasing concentrations of TNF alone or TNF + CHX for 24 hr and their viability was analyzed using the WST-1 assay (*Figure 9A*). Although TNF + CHX treatment strongly induced the death of Sharpin-deficient keratinocytes, TNF treatment alone had a very small effect in reducing the viability of Sharpin-deficient keratinocytes by about 10% compared with controls. Interestingly, the combined lack of FADD and RIPK3 in *Sharpin$^{cpdm/cpdm}$*;FADD$^{E-KO}$;*Ripk3$^{-/-}$* keratinocytes fully rescued the increased sensitivity of Sharpin-deficient keratinocytes to TNF + CHX (*Figure 9A*). However, keratinocytes obtained from *Sharpin$^{cpdm/cpdm}$*;*Fadd$^{fl/fl}$*;*Ripk3$^{-/-}$* showed a similar response to TNF + CHX as *Sharpin$^{cpdm/cpdm}$* keratinocytes, demonstrating that RIPK3 deficiency does not prevent the death of Sharpin-deficient keratinocytes. Therefore, Sharpin deficiency primarily sensitizes keratinocytes to FADD-mediated apoptosis and not to RIPK3-mediated necroptosis. To further examine a direct role of FADD in Sharpin-deficient cells without involvement of RIPK3, we generated HaCaT cells in which Sharpin and FADD were both stably knocked down by shRNA (*Figure 9B*). Upon treatment with TNF alone or TNF + CHX, HaCaT cells lacking both Sharpin and FADD showed reduced caspase-8 activity compared with Sharpin-deficient HaCaT cells (*Figure 9C,D*). Similar to the keratinocytes, we generated FADD-deficient *Sharpin$^{cpdm/cpdm}$* MEFs and analyzed the effect of FADD deficiency on apoptosis induced by TNF alone and by TNF + CHX in *Sharpin$^{cpdm/cpdm}$* MEFs (*Figure 9—figure supplement 1A*) and observed that FADD deficiency significantly suppressed the annexin V positive cells and caspase-8 activity in *Sharpin$^{cpdm/cpdm}$* MEFs (*Figure 9—figure supplement 1B and C*). To address an involvement of TRADD in TNF-induced sensitivity of Sharpin-deficient HaCaT cells, we used HaCaT cells which were knockdown for Sharpin and TRADD expression. In comparison to caspase-8 activity induced by TNF alone or TNF + CHX in Sharpin-deficient HaCaT cells, TRADD deficiency significantly suppressed caspase-8 activation (*Figure 9E–G*). These data collectively suggest that regulation of Sharpin-dependent anti-apoptosis signaling depends on FADD and TRADD in a cell-intrinsic manner.

## Discussion

Ubiquitination regulates a wide variety of biological functions by generating ubiquitin chains with different linkages on substrates (*Ikeda and Dikic, 2008*; *Komander and Rape, 2012*). One of the atypical linkage types is the Met-1/linearly-linked ubiquitin chain which is specifically generated by the LUBAC E3 ligase complex (*Kirisako et al., 2006*). Sharpin is a critical component of the LUBAC complex (*Gerlach et al., 2011*; *Ikeda et al., 2011*; *Tokunaga et al., 2011*) and its deficiency in mice leads to severe TNF-dependent inflammation in multiple organs including the skin (*Seymour et al., 2007*), suggesting that Sharpin has an important role in preventing inflammation. However, the mechanisms

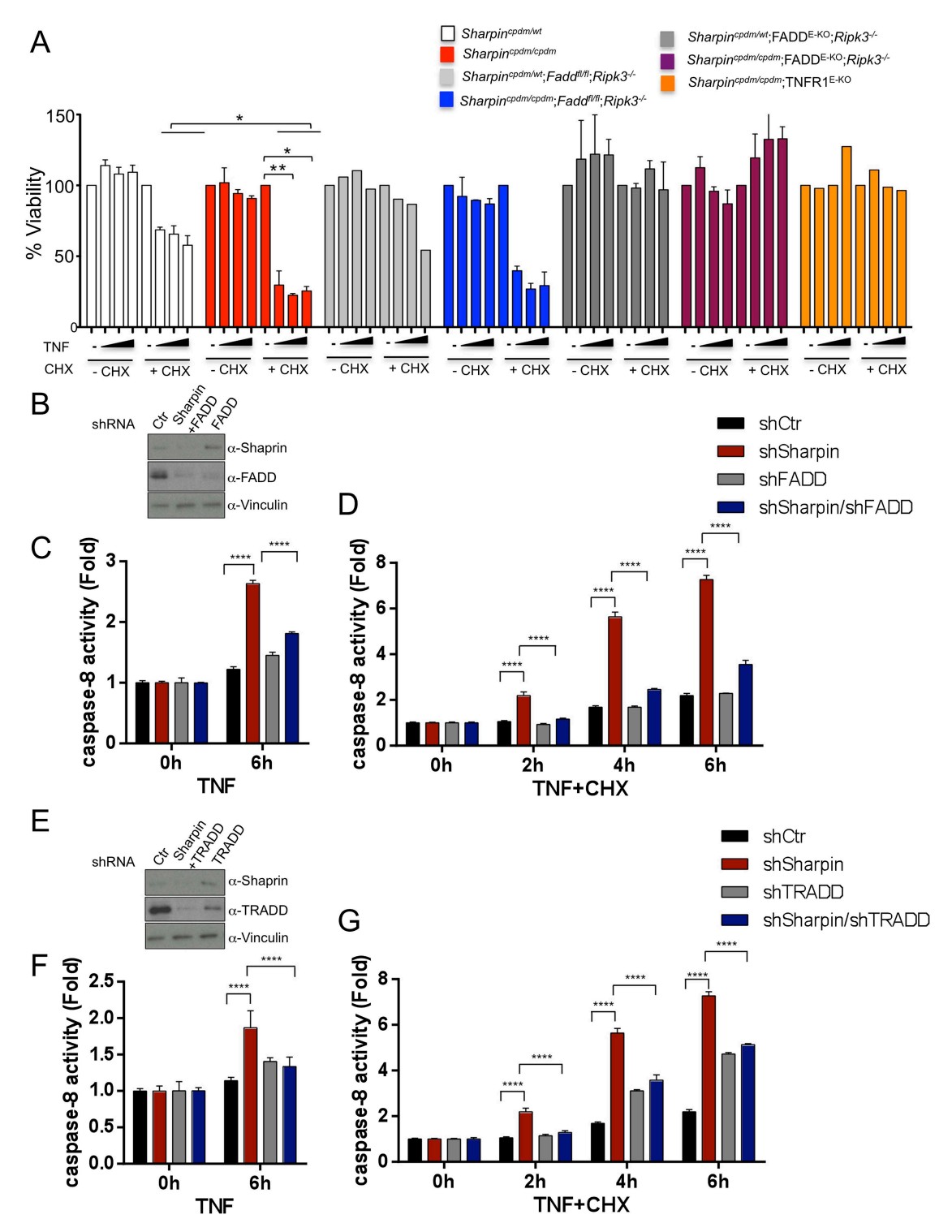

**Figure 9**. Fas-associated protein with death domain (FADD)- and tumor necrosis factor receptor 1-associated death domain (TRADD)-dependent enhanced sensitivity of Sharpin-deficient keratinocytes to tumor necrosis factor (TNF)-induced apoptosis. (**A**) Percentage viability of primary keratino-cytes isolated from *Sharpin^cpdm/cpdm*, *Sharpin^cpdm/cpdm*;*Ripk3^−/−*, *Sharpin^cpdm/cpdm*;FADD^E-KO;*Ripk3^−/−* and *Sharpin^cpdm/cpdm*;TNFR1^E-KO, and their respective control pups (n = 2) upon treatment with increasing TNF concentration (20, 50 and 100 ng/ml) in the presence or absence of cycloheximide (CHX) (1 μg/ml) for 24 hr. Viability of TNF-treated cells was normalized over their untreated control cells, and viability of TNF + CHX-treated cells was normalized over their CHX-treated control cells. The result shown here is representative of two independent experiments. The percentage viability was assessed using the

*Figure 9. Continued on next page*

*Figure 9. Continued*

WST-1 assay. Bars represent mean values ± SEM. Statistical significance was determined using the Student's *t* test (**p ≤ 0.01, *p ≤ 0.05). (**B–G**) Sharpin with FADD (**B**) or TRADD (**E**) was stably knocked down in HaCaT cells. Knockdown efficiency was analyzed by immunoblotting using α-Sharpin, α-FADD, and α-TRADD antibodies. Caspase-8 activity measurement in Sharpin knockdown, FADD knockdown, and double knockdown of Sharpin and FADD HaCaT cells (**C** and **D**) as well as Sharpin knockdown, TRADD knockdown, and double knockdown of Sharpin and TRADD HaCaT cells (**F** and **G**) upon treatment with TNF alone (**C** and **F**) or TNF + CHX (**D** and **G**) for the indicated time. Results are expressed as mean values ± SD. Statistical significance was determined using ANOVA test (****p ≤ 0.0001).

The following figure supplement is available for figure 9:

**Figure supplement 1**. Fas-associated protein with death domain (FADD) plays an important role in the Sharpin-dependent apoptosis signaling.

by which Sharpin prevents TNF-mediated inflammation in the skin and other organs have remained elusive.

We show here that TNFR1 signaling in keratinocytes is essential for the pathogenesis of skin inflammation in *Sharpin^cpdm/cpdm* mice. In addition, we provide genetic evidence that TNFR1-mediated, TRADD- and FADD-dependent apoptosis of Sharpin-deficient keratinocytes induces skin inflammation in these mice. RIPK3 deficiency only mildly delayed and ameliorated the severity of skin lesions in *Sharpin^cpdm/cpdm* mice, showing that RIPK3-dependent necroptosis plays a minor role in driving skin inflammation in this model, which was also shown by Rickard et al. (*Rickard et al., 2014*). These results demonstrate that Sharpin deficiency triggers skin inflammation by sensitizing keratinocytes to TNF-induced apoptosis. Furthermore, we found that inflammation in other organs including lung and liver as well as splenomegaly and altered splenic structure observed in *Sharpin^cpdm/cpdm* mice also depend on TNFR1 signaling and occur independently of the skin lesions. Together, these findings identified a cell-intrinsic function of Sharpin in inhibiting TNFR1-induced apoptosis that is essential for the maintenance of tissue homeostasis and the prevention of multi-organ inflammation.

In this study we have shown that Sharpin and HOIP, components of LUBAC, negatively regulate apoptotic pathways in keratinocytes. Our data suggest that Sharpin and HOIP act in an active ligase complex and raise the intriguing possibility that linear ubiquitination of an unknown target(s) inhibits TNF-induced apoptosis in keratinocytes. Ubiquitin signals have been implicated in the regulation of the death-inducing signaling complex (DISC). For example, cullin 3-based ubiquitination of caspase-8 by both Lys48- and Lys63-linked polyubiquitin chains brings caspase-8 into p62-containing aggregates leading to its activation and to commitment to apoptotic cell death (*Jin et al., 2009*). Furthermore, it was shown that FADD ubiquitination induced by Makorin Ring Finger Protein 1 (MKRN1) E3 ligase leads to the proteasome-dependent degradation of FADD, and MKRN1 depletion in breast cancer cells accelerates TNF-related apoptosis-inducing ligand (TRAIL)-induced DISC formation and apoptosis (*Lee et al., 2012*). Since we did not detect significant changes in total FADD protein levels in *Sharpin^cpdm/cpdm* MEFs in comparison to the wild type cells (*Figure 9—figure supplement 1A*), it suggests that protein stability is not controlled by Sharpin. It is also important to elucidate spatial and temporal regulation of DISC and TNFR complex II component ubiquitination and to identify the E3 ligases that mediate these ubiquitinations for the better understanding of biological functions of TNF signaling. Lastly, our study sheds new light on the specific functions of Sharpin as an integral component of the LUBAC E3 ligase complex. For example, both HOIL-1L and Sharpin are expressed in keratinocytes and have redundant roles in activating the NF-κB signaling pathway in MEFs (*Haas et al., 2009*; *Gerlach et al., 2011*). Yet, Sharpin-deficient mice develop inflammation in the skin whereas HOIL-1L knockout mice showed no obvious dermatitis phenotypes (*Tokunaga et al., 2009*). Interestingly, previous studies have shown that the analysis of complex formation of Sharpin, HOIL-1L, and HOIP suggested that some populations of Sharpin and HOIL-1L may exist in different molecular complexes from HOIP (*Gerlach et al., 2011*; *Tokunaga et al., 2011*). This raises the question whether Sharpin and HOIL-1L could individually regulate biological functions depending on the cell types, tissues, and pathogenic conditions. Precise examination of the effects of tissue-specific depletion of the LUBAC components in mice is needed to better understand the functional roles of each of the LUBAC components in vivo. Here we report the specific function of Sharpin in the regulation of apoptosis and skin inflammation, which are mediated through FADD and TRADD. Together, further studies of the regulatory mechanisms controlling inflammation in *Sharpin^cpdm/cpdm* mice will be important for the better understanding of the unique functions of Sharpin in vivo.

## Materials and methods

### Mice

The following mouse lines were used: *Sharpin$^{cpdm/cpdm}$* C57BL/KaLawRij (*Seymour et al., 2007*; *Ikeda et al., 2011*), *Tradd$^{fl/fl}$*, *Fadd$^{fl/fl}$* (*Mc Guire et al., 2010*), *K14-Cre* (*Pasparakis et al., 2002*), *Tnfrsf1a$^{-/-}$* (*Pfeffer et al., 1993*), *Tnfrsf1a$^{fl/fl}$* (*Van Hauwermeiren et al., 2013*), and *Ripk3$^{-/-}$* (*Newton et al., 2004*). All animal procedures were conducted in accordance with European, national, and institutional guidelines and protocols and were approved by local government authorities.

### Plasmids, antibodies and cells

pLKO.1-shRNA-control (AACAAGATGAAGAGCACCAACTCGAGTTGGTGCTCTTCATCTTGTT), pLKO.1-shRNA-mouse FADD (CCACACTTGGAGCCCAATAAACTCGAGTTTATTGGGCTCCAAGTGTGG), pLKO.1-shRNA-human HOIP (TGCTCCTTTGGCTTCATATATCTCGAGATATATGAAGCCAAAGGAGCA), pLKO.1-shRNA-human Sharpin (GTGTTCTCAGAGCTCGGTTTCCTCGAGGAAACCGAGCTCTGAG AACAC), pLKO.1-shRNA-human FADD (CATGGAACTCAGACGCATCTACTCGAGTAGATGCGTCTGA GTTCCATG), and pLKO.1-shRNA-human TRADD (CTGAAACTCCACTTGGCCTATCTCGAGATAGGC CAAGTGGAGTTTCAG) were generated by a standard subcloning method. The following antibodies were purchased and used according to the manufacturers' recommendations: anti-cleaved caspase-3 antibody (Asp175) (clone 5A1E; Cell Signaling Technology, Danvers, MA), anti-Vinculin antibody (Sigma, St Louis, MO), anti-PARP antibody (#9542; Cell Signaling Technology), anti-FADD antibody (clone 1F7; ENZO Life Sciences, Farmingdale, NY), anti-HOIP antibody (Aviva Systems Biology, San Diego, CA), anti-TRADD antibody (Santa Cruz Biotechnology, Santa Cruz, CA). Anti-Sharpin antibody has been described previously (*Tokunaga et al., 2009*; *Ikeda et al., 2011*).

Human Embryonic Kidney (HEK) 293T cells (ATCC, Boulevard Manassas, VA), immortalized mouse embryonic fibroblasts (MEFs), and HaCaT cells (a kind gift from Máté Borsos) were maintained at 37°C in 5% $CO_2$ condition in DMEM (Sigma) supplemented with 10% fetal calf serum (Life Technologies, Carlsbad, CA) and 100 U/ml penicillin and streptomycin (Invitrogen, Carlsbad, CA). Murine TNF was purchased from PeproTech (#315-01A, Rocky Hill, NJ).

### Immunohistochemistry and flow cytometry

Tissue samples from 12–18-week-old mice were fixed in 3.8–10% buffered formalin (skin) or 4% PFA (liver, lung and spleen) and subjected to histological analysis by H&E staining, TUNEL or immunohistochemical analysis. Slides were scanned using Mirax Slide Scanner (Carl Zeiss, Germany). The following antibodies were used: K14, K6, K10 and Loricrin (Covance, Prinston, NJ), F4/80 (clone A3-1, AbD Serotec, homemade), active caspase-3 (Cell Signaling Technologies), and TRADD (H-278, Santa Cruz). Secondary antibodies were coupled to Biotin (Dako, Germany); signal was amplified by avidin-biotin-HRP detection system (ABC VectorLab Elite Kit, Burlingame, CA) and detected by peroxidase substrate (VectorLab NovaRed). Sections were counterstained with hematoxylin for nuclei visualization. TUNEL staining was performed using TUNEL staining kit form Promega (Madison, WI), according to the manufacturer's instructions. Flow cytometric (FACS) analysis was performed on keratinocytes isolated from newborn pups as described previously (*Tscharntke et al., 2007*) and incubated with APC-conjugated anti-TNFR1 antibody (Biolegend, San Diego, CA) in PBS-BSA buffer, fixed in 4% PFA, followed by acquisition and analysis using FACS Calibur with accompanying software CellQuest (BD Bioscience, San Jose, CA).

### Apoptosis and cell death assays

For immunoblotting, the method is described elsewhere (*Ikeda et al., 2011*). Briefly, MEFs (0.05 × 10$^6$) were plated on 24-well plates. After 24 hr of subculturing, cells were treated with cycloheximide (CHX) (1 μM) (#C7698; Sigma) or TNF (10 ng/ml) (PeproTech) for the indicated times. Depending on the experimental set-ups, retroviral infection was combined. After the treatment, cells were harvested for SDS-PAGE followed by Western blot analysis. For the FACS analysis (Canto, BD Bioscience, San Jose, CA), the percentage of apoptotic cells was quantified by coupled annexin V antibody (#556419; BD Bioscience) staining and PI uptake. For WST-1 assay on MEFs, 9000 MEFs from wild type and TRADD-deficient mice were seeded in 96-well plates and treated with CHX (10 μg/ml), Necrostatin-1 (25 μM), and Z-VAD-FMK (20 μM) alone or in combination for 20 hr followed by incubation with WST-1 reagent (Roche, Indianapolis, IN) and measurement as per the manufacturer's instructions. For WST-1 assay on keratinocytes, keratinocytes were isolated as described previously (*Kumari et al., 2013*)

and seeded 20,000 cells/well from the indicated genotypes in 96-well plates and stimulated with TNF (20, 50 and 100 ng/ml) in the presence or absence of CHX (1 μg/ml) for 24 hr followed by incubation with WST-1 reagent (Roche) and measurement as per the manufacturer's instructions. For the caspase-8 activation assay, 50,000 cells/well in a 96-well plate were plated and treated with mTNF, CHX, or Necrostatin-1 for the indicated time. Lysates were used for the determination of caspase-8 activity in a luminescent signal-dependent manner following the manufacturer's protocol (Promega, caspase-Glo 8 Assay Systems).

### Immunofluorescence

Cells were fixed in 4% paraformaldehyde, permeabilized in 0.1%Triton/PBS, blocked in NGS/BSA/0.05%-Triton in PBS and incubated with α-cleaved caspase-3 antibody (Cell Signaling Technologies) followed by incubation with Alexa488 coupled anti-rabbit antibody (Invitrogen). Nuclei were stained by Vectashield mounting media with DAPI (VectorLabs).

### Lentiviral production

Lentiviral production and infection was performed according to the Addgene's pLKO.1 protocol with a minor modification. Briefly, pLKO.1 vector with a packaging and envelope plasmids were transfected into HEK293T cells using Gene Juice (Novagen). After 36 hr of transfection, released lentivirus particles were filtered and used for infection of target MEFs or HaCaT cells using polybrene (4 μg/ml). After 48 hr of infection, cells were selected with puromycin (2 μg/ml). Expression of infected protein was monitored by western blotting.

### Statistical analysis

Statistical significance was determined using ANOVA (one-way or two-way) and unpaired or paired Student's $t$ test (two-tailed) by Prism 6 software (Graph Pad) or Microsoft Excel (*$p \leq 0.05$, **$p \leq 0.01$, ***$p \leq 0.001$, ****$p \leq 0.0001$).

## Acknowledgements

We would like to acknowledge the HistoLab, the Biooptics, the molecular biology service, the animal house (IMP/IMBA, Vienna, Austria) and C Uthoff-Hachenberg, J Buchholz, E Mahlberg, B Kühnel, D Beier, B Hülser and E Stade for technical support. We also thank all the laboratory members of the Ikeda and Pasparakis groups for active discussion on the project and comments on the manuscript.

## Additional information

### Competing interests

ID: Reviewing editor, *eLife*. The other authors declare that no competing interests exist.

### Funding

| Funder | Grant reference number | Author |
|---|---|---|
| Austrian Science Fund | P25508 | Fumiyo Ikeda |
| European Research Council | 250241 | Ivan Dikic |
| LOEWE Zentrum AdRIA | Ub-Net | Ivan Dikic |
| Deutsche Forschungsgemeinschaft | SFB670 | Manolis Pasparakis |
| European Commission | 223404 | Manolis Pasparakis |
| Deutsche Krebshilfe | 110302 | Manolis Pasparakis |
| Else Kröner-Fresenius-Stiftung | | Manolis Pasparakis |
| Helmholtz Association | Helmholtz Alliance Preclinical Comprehensive Cancer Center | Manolis Pasparakis |
| Human Frontier Science Program | | Jaime Lopez-Mosqueda |
| European Research Council | 2012-ADG_20120314 | Manolis Pasparakis |
| European Commission | 223151 | Manolis Pasparakis |

| Funder | Grant reference number | Author |
|---|---|---|
| Deutsche Forschungsgemeinschaft | SFB829 | Manolis Pasparakis |
| Deutsche Forschungsgemeinschaft | SPP1656 | Manolis Pasparakis |

The funders had no role in study design, data collection and interpretation, or the decision to submit the work for publication.

## Author contributions

SK, YR, JL-M, Acquisition of data, Analysis and interpretation of data, Drafting or revising the article; RS, MR, SL, JK, CM, Acquisition of data, Analysis and interpretation of data; ID, MP, Conception and design, Drafting or revising the article; FI, Conception and design, Acquisition of data, Analysis and interpretation of data, Drafting or revising the article

## Ethics

Animal experimentation: All animal procedures were strictly conducted in accordance with the Recommendations and Regulations of the 'Society for Laboratory Animal Science- GV-SOLAS' and of the 'Federation of European Laboratory Animal Science Associations - FELASA'. All of the animals were handled according to the guidelines and approved protocols of 1) European guideline: EUROPEAN CONVENTION FOR THE PROTECTION OF VERTEBRATE ANIMALS USED FOR EXPERIMENTAL AND OTHER SCIENTIFIC PURPOSES, Tierversuchs-Richtlinie (EC) EUROPEAN COUNCIL DIRECTIVE of 24 November 1986 on the approximation of laws, regulations and administrative provisions of the Member States regarding the protection of animals used for experimental and other scientific purposes (86/609/EEC), 2) national guidelines: Governing Austrian or German laws, 3) institutional guidelines and 4) approved institutional protocols by local government authorities (IMBA and Cologne University).

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
