## [Decision Letter]

Thank you for sending your work entitled “Sharpin prevents TNFR1-induced keratinocyte apoptosis and skin inflammation via linear ubiquitination of FADD” for consideration at *eLife*. Your article has been favorably evaluated by Tadatsugu Taniguchi (Senior editor) and 3 reviewers, one of whom is a member of our Board of Reviewing Editors.

The Reviewing editor and the other reviewers discussed their comments before we reached this decision, and the Reviewing editor has assembled the following comments to help you prepare a revised submission.

In this manuscript, Kumari et al. employ a variety of genetically modified mice and nicely demonstrate the FADD and TRADD-mediated TNF-induced apoptosis in epidermal keratinocytes is mainly responsible for skin inflammation and dermatitis in Sharpin-deficient cpdm mice. They also show that RIP3-mediated necroptosis only plays a minor role in the skin pathology, but the absence of TNFR1 and RIP3 can significantly (but not completely) block liver and lung inflammation as well as splenomegaly in Sharpin-deficient mice. Secondly, they show that Sharpin-deficient MEF cells are more sensitive to TNF induction of apoptosis and FADD appears to play an important role in this process. Thirdly, the rest of the manuscript tries to suggest that linear ubiquitination of FADD by Sharpin and LUBAC negatively regulates TNF-induced apoptosis, which may be the mechanism underlying the increase sensitivity of Sharpin deficiency to TNF-induced apoptosis. Overall, the first part of the story provides strong data and important insights into the mechanism of skin and multiple organ inflammation observed in cpdm mice. These, together with the second part of the story, nicely demonstrate the physiological function of Sharpin and likely also the LUBAC complex in negatively regulating apoptosis downstream of the death receptors. The third part aimed at providing a thorough biochemical mechanism for the apoptosis inhibition function of Sharpin is rather weak and in fact not essential for the story that is already pretty informative. Therefore, the authors should consider addressing the following comments when revising their manuscript.

Major comments:

1) For Figures 2, 4 and 5, the authors should include littermate mice as the appropriate controls.

2) The authors provide strong in vitro data that LUBAC can ubiquitinate FADD (Figure 9) and that Sharpin but not HOIL-1l/HOIP can bind to FADD directly. However, the functional data supporting that ubiquitination of FADD by LUBAC suppresses TNF-induced apoptosis is rather weak, indirect and not sufficiently convincing. The most important piece of data is the FADD K0 mutant rescue assay in Figure 9, in which the difference of apoptosis induction (PARP and caspase-3 cleavage) between WT and FADD K0 complementation is very subtle and not sufficient to justify the conclusion. The FADD-ubiquitin chimera experiment is difficult to interpret and should be removed from the manuscript, particularly in the absence of observing FADD ubiquitination in vivo.

The manuscript contains sufficient new information from Figures 1, 2, 3, 4, 5, 6 and 7; starting from characterizing the phenotype using various genetically modified mice to the demonstration that FADD is responsible for the increased apoptosis sensitivity to TNF stimulation. There is no need of defining the substrate of LUBAC involved in this process. If the authors really want to make the point here, they need to show in vivo linear ubiquitination of endogenous FADD upon TNF stimulation, ideally in keratinocytes, and this ubiquitination is absent in Sharpincpdm/cdpm cells.

3) In view of the recent paper by Berger et al (JI, June 2014), cell death observed in the Sharpin deficient mice/cells could be expected to be RIPK1 kinase dependent. Therefore, the authors should use another cell model rather than the TNF/CHX treatment in MEFs, since this kills in a RIPK1 kinase-independent way and may not reflect the in vivo situation. In fact, keratinocytes are much more appropriate, for demonstrating that increased apoptosis in the skin is key to the development of skin inflammation and dermatitis in cpdm mice. The authors should examine whether TNF is sufficient to induce apoptosis in primary keratinocytes derived from cpdm mice and compare the cell death of WT keratinocytes. The authors then can further examine whether FADD and TRADD (individually or double) deficiency can block the sensitivity of Sharpincpdm/cdpm keratinocytes. This can either be done by using the mice they already have or siRNA knockdown of FADD and TRADD. The authors argued that Sharpin prevents apoptosis independently from its role in NF-kB pathway and this should be further examined other relevant cellular systems in which the Sharpin deficient cells are killed by TNF without adding CHX.

4) At the cellular level, Sharpin-deficiency sensitizes the cells to TNF-induced apoptosis. The authors should examine whether the apoptosis is RIPK1 kinase-dependent. They can also look at the RIPK1-containing complexes and compare with that in the presence of Sharpin.

5) Sharpin is a component of the LUBAC complex. Another important question that the authors should address is whether Sharpin really functions through LUBAC in inhibiting TNF-induced apoptosis. Therefore, the authors should examine whether other LUBAC components are involved in the observed cellular apoptosis phenotypes, ideally in keratinocytes.

---

## [Author Response]

*1) For*
Figures 2, 4 and 5*, the authors should include littermate mice as the appropriate controls*.

Due to the complex breeding schemes required for the generation of some of the mouse lines described, it was not always possible to obtain littermates with all possible control genotypes. Whenever possible, we have made an effort to include littermate controls, but this was not always feasible. However, we would like to point out that the variability we observed in the phenotype of the cpdm mice was also seen within litters. Even in breeding of cpdm heterozygous mice without any additional alleles we often observed that littermate *Sharpin*
^*cpdm/cpdm*^ mice exhibited variability in the onset of the skin lesions, with some mice starting to show lesions as early as at 9 weeks of age and some as late as 12 weeks. Because of this variability in the onset of the phenotype even within the same litter, we have selected representative mice to show for each genotype that correspond to the average stage of disease observed among a large number of individual animals examined.

Specifically, we used the following controls for each of the lines:

Figure 1.

- *Sharpin*^*cpdm/cpdm*^/*Tnfr1f*^*l/fl*^*/K14Cre* line: For this line, we have used littermates that are cpdm/cpdm, *Tnfr1*^fl/fl^ but not expressing the K14Cre transgene. Littermate control for this line is added in Figure 1.

- *Sharpin*^*cpdm/cpdm*^ /*Tnfr1*^*-/-*^ line: By breeding double heterozygous mice the probability of obtaining *Sharpin*^*cpdm/cpdm*^/*Tnfr1*^wt/wt^ or *Sharpin*^*cpdm/cpdm*^/*Tnfr1*^-/-^ mice is 1/16. By considering an average number of 6-7 pups per litter, the probability to have one *Sharpin*^*cpdm/cpdm*^ /*Tnfr1*^wt/wt^ and one *Sharpin*^*cpdm/cpdm*^/*Tnfr1*^-/-^ mouse within the same litter is very low. In order to obtain the number of mice we needed for a comprehensive analysis of the phenotype we also used breeding of heterozygous cpdm mice that were homozygous *Tnfr1*^-/-^ and compared the resulting mice to *Sharpin*^*cpdm/cpdm*^ or *Sharpin*^*cpdm/cpdm*^/*Tnfr1f*^*l/fl*^ mice from other breedings.

- *Sharpin*^*cpdm/cpdm*^ /*Fadd*^*fl/fl*^;*Ripk3*^*-/-*^ line: For this line we compared littermates that are K14Cre positive (*Sharpin*^*cpdm/cpdm*^/FADD^E-KO^/*Ripk3*^*-/-*^) that are FADD-deficient in keratinocytes with K14Cre negative (*Sharpin*^*cpdm/cpdm*^/*Fadd*^*fl/fl*^/*Ripk3*^*-/-*^) that express FADD in keratinocytes.

Therefore, we compared the effect of double FADD/RIPK3 knockout to that of single RIPK3 knockout. In addition, we have now added littermate controls for *Sharpin*^*cpdm/cpdm*^ /*Fadd*^*fl/fl*^;*Ripk3*^*-/-*^ line (*Sharpin*^*wt/wt*^/*Fadd*^*fl/wt*^;*Ripk3*^*-/-*^) in Figure 2. To have the best possible matched controls, we have replaced some of the pictures of the old figures. In Figure 5
*(left most panel),* histology of liver and spleen has been replaced. In Figure 5, the macroscopic spleen picture has been replaced with spleen picture of littermate controls. Wherever possible, in the new version of the figures or figure legends, we have stated the exact genotype of the mouse used in each figure.

*2) The authors provide strong in vitro data that LUBAC can ubiquitinate FADD (*Figure 9*) and that Sharpin but not HOIL-1l/HOIP can bind to FADD directly. However, the functional data supporting that ubiquitination of FADD by LUBAC suppresses TNF-induced apoptosis is rather weak, indirect and not sufficiently convincing. The most important piece of data is the FADD K0 mutant rescue assay in*
Figure 9*, in which the difference of apoptosis induction (PARP and caspase-3 cleavage) between WT and FADD K0 complementation is very subtle and not sufficient to justify the conclusion*.

We agree with the reviewers and removed FADD-ubiquitination data, interaction assay of FADD and LUBAC components, and FADD K0 data and the related panels.

*The FADD-ubiquitin chimera experiment is difficult to interpret and should be removed from the manuscript, particularly in the absence of observing FADD ubiquitination in vivo*.

We agree with the reviewers and removed FADD-ubiquitin chimera data.

*The manuscript contains sufficient new information from*
Figures 1, 2, 3, 4, 5, 6 and 7*; starting from characterizing the phenotype using various genetically modified mice to the demonstration that FADD is responsible for the increased apoptosis sensitivity to TNF stimulation. There is no need of defining the substrate of LUBAC involved in this process. If the authors really want to make the point here, they need to show in vivo linear ubiquitination of endogenous FADD upon TNF stimulation, ideally in keratinocytes, and this ubiquitination is absent in Sharpincpdm/cdpm cells*.

We thank the reviewers for this suggestion and reformatted the manuscript by focusing on the role of FADD in TNF-mediated apoptosis of sharpin-deficient cells without its ubiquitination. Therefore, panels which related to FADD ubiquitination, in which linear-ubiquitin chain interacter ABIN1 was tested were removed.

*3) In view of the recent paper by Berger et al (JI, June 2014), cell death observed in the Sharpin deficient mice/cells could be expected to be RIPK1 kinase dependent. Therefore, the authors should use another cell model rather than the TNF/CHX treatment in MEFs, since this kills in a RIPK1 kinase-independent way and may not reflect the in vivo situation. In fact, keratinocytes are much more appropriate, for demonstrating that increased apoptosis in the skin is key to the development of skin inflammation and dermatitis in cpdm mice. The authors should examine whether TNF is sufficient to induce apoptosis in primary keratinocytes derived from cpdm mice and compare the cell death of WT keratinocytes*.

We agree with the reviewers that addressing the role of Sharpin in TNF-induced death of keratinocytes is more relevant to the *in vivo* studies. To assess TNF-induced death we prepared primary keratinocytes lacking sharpin, sharpin and RIPK3, sharpin and TNFR1 and sharpin, RIPK3 and FADD as well as control cells from littermates as shown in Figure 9. We treated these primary keratinocytes with increasing concentration of TNF (20, 50, 100ng/ml) alone or together with cycloheximide (CHX) (1 µg/ml) and analyzed their viability 24 hours after stimulation using the WST-1 assay. We found that in the presence of CHX, TNF induces the death of about 80% of sharpin-deficient keratinocytes compared to about 40% of control cells (new Figure 9). TNF+CHX-induced death was not inhibited by RIPK3 deficiency but was fully prevented by lack of TNFR1 or double FADD/RIPK3 deficiency, showing that in the presence of CHX, sharpin deficiency sensitizes keratinocytes to TNF induced FADD-dependent apoptosis. However, in the absence of CHX, we observed that TNF treatment (only at the higher concentration 50 and 100 ng/ml) reduced the viability of *Sharpin*^*cpdm/cpdm*^ keratinocytes by about 8-10% compared to heterozygous or wild type (*Sharpin*^*cpdm/wt*^ or *Sharpin*^*wt/wt*^ ) control kearatinocytes. This effect of TNF was too weak to provide robust data allowing the comparison between the different genotypes, as a reduction of viability by 8-10% is within the range of the normal variability observed in this assay using primary keratinocytes. We conclude that while TNF clearly induces apoptosis of sharpin-deficient keratinocytes *in vivo* that is responsible for skin inflammation in cpdm mice, in primary keratinocyte cultures TNF stimulation does not induce considerable amount of cell death in the absence of CHX to allow reliable quantification of the effect. It is possible that the *in vivo* context and the keratinocyte differentiation status are responsible for the difference between the epidermal keratinocytes *in vivo* and the primary cultured cells.

To bypass this problem, in parallel we performed experiments in a human keratinocyte cell line, HaCaT cells, in the presence or absence of Sharpin by knockdown. We observed that TNF alone induced mildly elevated apoptosis in Sharpin-knockdown HaCaT cells compared to control cells. Although the increase in cell death was in the range observed for the primary cells, we were able to quantify this difference reliable using Annexin V staining and measurement of caspase-8 activity (new Figure 8). These results are similar to what we observed before in Sharpin deficient MEFs (new Figure 7), and this effect was further enhanced in the presence of CHX. To address the role of RIPK1 kinase activity, we treated cells with TNF in the presence of the RIPK1 kinase inhibitor Necrostatin -1 (Nec-1). As shown in new Figure 8, in the presence of Nec-1, we observed reduction of caspase-8 activity in sharpin-deficient cells. This result shows a partial inhibition of TNF-induced apoptosis in the presence of Nec-1, suggesting a role of RIPK1 in this signaling pathway as indicated by the *in vivo* result from Berger et al. (2014).

*The authors then can further examine whether FADD and TRADD (individually or double) deficiency can block the sensitivity of Sharpincpdm/cdpm keratinocytes. This can either be done by using the mice they already have or siRNA knockdown of FADD and TRADD. The authors argued that Sharpin prevents apoptosis independently from its role in NF-kB pathway and this should be further examined other relevant cellular systems in which the Sharpin deficient cells are killed by TNF without adding CHX*.

To examine whether FADD and TRADD deficiency can block the TNF-induced sensitivity of Sharpin-deficient HaCaT cells, we generated knockdown HaCaT cells lacking Sharpin+FADD or Sharpin+TRADD and examined TNF -induced apoptosis with and without CHX (new Figure 9). We observed that both FADD and TRADD deficiencies suppressed TNF-induced apoptosis (with or without CHX) in Sharpin knockdown HaCaT cells, which further indicates that enhanced sensitivity of sharpin-deficient keratinocytes to TNF is FADD and TRADD- dependent. It was an identical case with FADD knockdown in Sharpin deficient MEFs (Figure 9).

As mentioned in our response to the previous comment, to address the role of FADD and RIPK3 in TNF-induced death of primary keratinocytes, we isolated primary keratinocytes from *Sharpin*^*cpdm/cpdm*^
*or Sharpin*^*cpdm/wt*^;*Fadd*^*fl/fl*^;*Ripk3*^-/-^ lines, stimulated with TNF alone or with TNF + CHX and analyzed their viability using the WST-1 assay. Interestingly, the increased sensitivity of *Sharpin*^*cpdm/cpdm*^ keratinocytes to TNF+CHX was fully rescued in *Sharpin*^*cpdm/cpdm*^;FADD^E-KO^;*Ripk3*^-/^ keratinocytes. However keratinocytes obtained from *Sharpin*^*cpdm/cpdm*^;*Fadd*^*fl/fl*^;*Ripk3*^-/-^ mice showed similar response to TNF+CHX as observed in *Sharpin*^*cpdm/cpdm*^ keratinocytes (new Figure 9). Therefore, double FADD/RIPK3 deficiency fully prevented TNF+CHX induced apoptosis of Sharpin-deficient keratinocytes while RIPK3 deficiency alone had no effect, confirming that TNF kills Sharpin-deficient keratinocytes primarily due to FADD-mediated apoptosis (new Figure 9). However, as discussed above, TNF treatment alone had a minor effect in inducing the death of Sharpin-deficient primary keratinocytes, which we could not quantify reliably to allow comparison between the different genotypes. Based on these results, we cannot reliably conclude whether the role of Sharpin in preventing TNF-induced apoptosis is independent of NF-κB activity. It is likely that Sharpin prevents TNF-induced apoptosis by both NF-κB dependent and independent activities.

*4) At the cellular level, Sharpin-deficiency sensitizes the cells to TNF-induced apoptosis. The authors should examine whether the apoptosis is RIPK1 kinase-dependent. They can also look at the RIPK1-containing complexes and compare with that in the presence of Sharpin*.

As we mentioned above (#3), based on the effect of RIPK1 inhibitor Nec-1 on TNF-treated Sharpin deficient HaCaT cells, RIPK1 is suggested to be involved in Sharpin-dependent apoptosis regulation in keratinocytes. We agree that to understand the RIPK1 complex with and without Sharpin in keratinocytes is important and interesting. Unfortunately, we could not observe the RIP1-containing complex with LUBAC components by performing co-IP using anti-RIP1 antibody, with TNF+zVAD-treated HaCaT cells, leaving out the question whether Sharpin affects the RIP1-complex formation.

*5) Sharpin is a component of the LUBAC complex. Another important question that the authors should address is whether Sharpin really functions through LUBAC in inhibiting TNF-induced apoptosis. Therefore, the authors should examine whether other LUBAC components are involved in the observed cellular apoptosis phenotypes, ideally in keratinocytes*.

We agree that it is an important point. To address this issue, we generated HOIP-knockdown HaCaT keratinocyte cell line using shRNA. TNF alone or TNF+CHX induced increased apoptosis of HaCaT cells (lacking HOIP), suggesting an important role of the LUBAC complex in the preventing TNF induced apoptosis in keratinocytes. New data are added in Figure 8—figure supplement 1.